



# Assimilating spaceborne lidar dust extinction improves dust forecasts

Jerónimo Escribano[1], Enza Di Tomaso[1], Oriol Jorba[1], Martina Klose[1,2], Maria Gonçalves Ageitos[1,3], Francesca Macchia[1], Vassilis Amiridis[4], Holger Baars[5], Eleni Marinou[4], Emmanouil Proestakis[4], Claudia Urbanneck[5], Dietrich Althausen[5], Johannes Bühl[5], Rodanthi-Elisavet Mamouri[6,7], and Carlos Pérez García-Pando[1,8]

[1]Barcelona Supercomputing Center (BSC), Barcelona, Spain
[2]Karlsruhe Institute of Technology (KIT), Institute of Meteorology and Climate Research (IMK-TRO), Department Troposphere Research, Karlsruhe, Germany
[3]Technical University of Catalonia (UPC), Barcelona, Spain
[4]National Observatory of Athens (NOA), Athens, Greece
[5]Leibniz Institute for Tropospheric Research (TROPOS), Leipzig, Germany
[6]Cyprus University of Technology, Cyprus
[7]ERATOSTHENES Center of Excellence, Cyprus
[8]ICREA, Catalan Institution for Research and Advanced Studies, Barcelona, Spain

**Correspondence:** Jerónimo Escribano (jeronimo.escribano@bsc.es)

**Abstract.** Atmospheric mineral dust has a rich tri-dimensional spatial and temporal structure that is poorly constrained in forecasts and analyses when only column-integrated aerosol optical depth (AOD) is assimilated. At present, this is the case of most operational global aerosol assimilation products. Aerosol vertical distributions obtained from space-borne lidars can be assimilated in aerosol models, but questions about the extent of their benefit upon analyses and forecasts along with their

consistency with AOD assimilation remain unresolved. Our study thoroughly explores the added value of assimilating space-borne vertical dust profiles, with and without the joint assimilation of dust optical depth (DOD). We also discuss the consistency in the assimilation of both sources of information and analyse the role of the smaller footprint of the space-borne lidar profiles upon the results. To that end, we have performed data assimilation experiments using dedicated dust observations for a period of two months over Northern Africa, the Middle East and Europe. We assimilate DOD derived from VIIRS/SUOMI-NPP

Deep Blue, and for the first time CALIOP-based LIVAS pure-dust extinction coefficient profiles on an aerosol model. The evaluation is performed against independent ground-based DOD derived from AERONET Sun photometers and ground-based lidar dust extinction profiles from field campaigns (CyCARE and Pre-TECT). Jointly assimilating LIVAS and Deep Blue data reduces the root mean square error (RMSE) in the DOD by 39% and in the dust extinction coefficient by 65% compared to a control simulation that excludes assimilation. We show that the assimilation of dust extinction coefficient profiles provides a

strong added value to the analyses and forecasts. When only Deep Blue data are assimilated the RMSE in the DOD is reduced further, by 42%. However, when only LIVAS data are assimilated the RMSE in the dust extinction coefficient decreases by 72%, the largest improvement across experiments. We also show that the assimilation of dust extinction profiles yields better skill scores than the assimilation of DOD under equivalent sensor footprint. Our results demonstrate the strong potential of





future lidar space missions to improve desert dust forecasts, particularly if they foresee a depolarization lidar channel to allow
discriminating desert dust from other aerosol types.

# 1  Introduction

The spatial and temporal distribution of atmospheric aerosol can be optimally estimated by combining observations and numerical models using data assimilation (DA) techniques. The resulting fields, referred to as aerosol analyses, serve as initial conditions for aerosol forecasting. Long-term and consistent analyses, so-called aerosol reanalyses, are useful for investigating
aerosol variability, trends, impacts and climate feedbacks, and they are produced with the same DA techniques (Benedetti et al., 2009; Lynch et al., 2016; Randles et al., 2017; Yumimoto et al., 2017; Inness et al., 2019; Di Tomaso et al., 2021).

A key uncertainty in current models is the representation of the aerosol vertical distribution (Pérez et al., 2006; Koffi et al., 2016; Benedetti et al., 2018; Konsta et al., 2018). Most operational aerosol forecast systems rely on the assimilation of column–integrated aerosol optical depth (AOD) from satellite-borne instruments (e.g. Xian et al., 2019). Consequently, the vertical
structure is mainly propagated from the numerical model and only slightly and indirectly from the assimilated observations. In the last decade, a few studies have investigated the assimilation of vertical aerosol profiles from LIDAR instruments, both satellite (e.g. CALIOP, Winker et al., 2010) and ground-based (e.g. EARLINET, Pappalardo et al., 2014), showing the potential of vertical profiling to improve the four-dimensional representation of aerosols in analyses (Sekiyama et al., 2010; Zhang et al., 2011; Wang et al., 2014; Kahnert and Andersson, 2017; Cheng et al., 2019) and forecasts (Zhang et al., 2011; Wang et al.,
2014). Difficulties preventing an effective assimilation of vertical profiles in operational settings include the poor coverage of ground-based observations, the narrow footprint of satellite observations, potential inconsistencies with other assimilated observations, and underrepresented forecasting uncertainty in the vertical, among other.

Our study focuses on the assimilation of desert dust aerosol lidar observations around the two most prolific source regions on Earth: Northern Africa and the Middle East. Dust models are subject to substantial uncertainties in the description of
lower boundary conditions relevant for dust emission, modelled wind speed, dust emission processes, vertical mixing, particle properties and deposition (Huneeus et al., 2011; Kok et al., 2020; Klose et al., 2021). Thus combining modelling with dust observations through DA is a powerful method to increase the quality of emission estimates (Escribano et al., 2016, 2017) and dust forecasts. We are specifically interested here on the impact of assimilating spaceborne lidar profiles upon dust forecasts. Dust is the largest continental contributor to the global aerosol load and impacts marine (Jickells et al., 2005) and land bio-
geochemistry (Okin et al., 2004), radiative fluxes (DeMott et al., 2009; Kok et al., 2017; Marinou et al., 2019), human health (Du et al., 2015) and economy (Kosmopoulos et al., 2018; Papagiannopoulos et al., 2020). Properly representing its vertical structure both within the planetary boundary layer over sources and in the free troposphere in the outflow areas is particularly important to predict its long range transport and associated impacts (O'Sullivan et al., 2020). Despite this important role, the dust vertical structure in models and forecasts is poorly constrained by observations (Benedetti et al., 2014). So far, only lidar
measurements, either from space or from ground, can deliver vertical profiles of the dust load.



Our work involves both modelling and data assimilation aspects, along with the handling of observations and their uncertainty. We use the Multiscale Online Non-hydrostatic AtmospheRe CHemistry (MONARCH) model, formerly known as NMMB/BSC-Dust (Pérez et al., 2011; Klose et al., 2021), enhanced with a Local Ensemble Transform Kalman Filter data assimilation capability (Di Tomaso et al., 2017). MONARCH provides dust forecasts at the WMO Sand and Dust Storm Warning Advisory and Assessment System (SDS-WAS) Regional Centers for Northern Africa, Middle East and Europe (http://sds-was.aemet.es/; http://dust.aemet.es/) hosted by the the Spanish Meteorological agency (AEMET) and the Barcelona Supercomputing Center. We address the challenge of how to best express model uncertainty also in the vertical coordinate, and consequently in the dust transport, generating an ensemble for MONARCH based on both meteorological and dust source perturbations. Rubin et al. (2016) showed that combining meteorology and aerosol source ensembles produce sufficient spread in outflow regions that positively impacts the results. Characterizing model uncertainty is key to effectively assimilate observations; spatial and multivariate structures of the error background covariance determine the spread of observational information in space and across variables, allowing for statistically consistent increments between neighbouring grid points, also along the vertical dimension. The use of an ensemble-based data assimilation scheme, such as the one used in this work, allows for background covariances to evolve with the forecast.

Assimilating dust in models is possible to the extent that there are dust-specific retrievals with suitable coverage, quality and uncertainty quantification. Progress has been made recently to provide dust products from satellite-borne spectroradiometers in the visible (e.g., Pu and Ginoux, 2016), IASI (e.g., Capelle et al., 2018; Clarisse et al., 2019), ground and satellite-based lidar instruments (Mamouri and Ansmann, 2014; Amiridis et al., 2015) or combinations of reanalyses with satellite retrievals (Gkikas et al., 2021). In our study we assimilate pixels with dust retrievals from the VIIRS Deep Blue AOD product (Hsu et al., 2019), along with LIVAS pure-dust extinction coefficient profiles from CALIPSO as described in Amiridis et al. (2013, 2015) and Marinou et al. (2017). Dust is irregularly shaped. Yet most models assume dust to be a perfect sphere. In contrast to most previous studies we assume spheroidal-shape dust particles in the observation operator that translates mass into dust extinction and dust optical depth (DOD). Finally our analyses and analysis-initialised forecasts are evaluated against independent observations, namely dust-filtered AOD from ground-based AERONET observations and LIDAR dust extinction coefficient profiles collected during the Pre-TECT (http://pre-tect.space.noa.gr) and CyCARE (Radenz et al., 2017) campaigns between the 19 and 23 of April 2017.

The paper is organized as follows. In the Section 2 we describe the data and methods employed in this study, including the observations used for assimilation (Section 2.1) and evaluation (Section 2.2), the modeling system (Section 2.3), and the data assimilation scheme and parameter settings (Section 2.4). The experimental setup and the performance scores used for the evaluation are described in Section 2.5. In Section 3 we investigate the potential improvements in the representation of the dust vertical structure by assimilating dust dedicated profiling information in a close-to-optimal data assimilation framework. We also assesses the overall benefit of applying constraints on both the dust total column extinction and the dust extinction profile. Finally, we compare vertically-resolved vs column-integrated data assimilation under comparable temporal and spatial geographical sampling. Section 4 concludes the paper highlighting the main results obtained.



## 2 Data and methods


We performed data assimilation experiments to evaluate the impact of assimilating satellite products of DOD and vertically-resolved dust extinction coefficient, either alone or in combination. These two datasets are described in Section 2.1. The experiments were evaluated against independent ground-based Sun photometer and lidar observations that are described in Section 2.2.

The modelling and data assimilation systems, described in Sections 2.3 and 2.4, respectively, were optimised in a number of aspects including the generation of ensemble perturbations, the spatial and temporal localisation that creates a smooth limit upon the observation influence in the analysis fields, and the optical properties used in the observation operator. A description of the experiments and their evaluation are provided in Section 2.5

### 2.1   Assimilated observations

### 2.1.1   CALIOP-based LIVAS dataset

Pure-dust profiles assimilated in this study were derived from the global 3-D ESA-LIVAS database (LIdar climatology of Vertical Aerosol Structure for space-based lidar simulation studies, Amiridis et al., 2013, 2015). LIVAS, developed based on multiyear CALIPSO (Cloud-Aerosol Lidar and Infrared Pathfinder Satellite Observations, Winker et al., 2009) CALIOP (Cloud–Aerosol Lidar with Orthogonal Polarization) observations, provides averaged profiles of aerosol and cloud optical

properties on a uniform 1°x1°grid resolution, for the CALIPSO-defined aerosol and cloud subtype classes. The methodology of LIVAS to decouple the pure-dust backscatter coefficient component from the total aerosol mixture is based on the one-step POLIPHON technique (POlarization-LIdar PHOtometer Networking, Tesche et al., 2009), established in the framework of EARLINET (European Aerosol Research Lidar Network, http://www.earlinet.org/, last access: 25 May 2021, Pappalardo et al., 2014). It uses CALIOP Version 4 (V4) Level 2 (L2) aerosol profiles of backscatter coefficient and particulate depolarization

ratio products at 532 nm. Moreover, the procedure applies several quality-assurance procedures (Tackett et al., 2018) and suitable geographically dependent dust lidar ratio conversion factors (e.g. 55 sr for Saharan Desert, 40 sr for Middle East) to obtain the atmospheric aerosol profiles of pure-dust extinction coefficient at 532 nm (Amiridis et al., 2013; Marinou et al., 2017; Proestakis et al., 2018) at CALIPSO per-orbit level, used in the present study.

Accordingly, prior to assimilation, the profiles of LIVAS pure-dust extinction coefficient at 532 nm were aggregated to

the horizontal resolution of the model. In the regridding process, error definitions and filtering of CALIOP profiles followed procedures similar to Cheng et al. (2019). More specifically, Cheng et al. (2019) used CALIOP optical products under the condition that at least 20 CALIPSO L2 profiles were provided in each 2° x 2° model grid cells. Considering the finer model grid resolution of 0.66° x 0.66° of the present study, in an analogous approach to Cheng et al. (2019), a threshold of at least 3 Quality Assured (QA) Cloud-Free (CF) CALIPSO L2 profiles were set, achieving similar proportion of horizontal geographical

coverage to Cheng et al. (2019). Similar was the filtering approach followed for the coefficient of variation (standard deviation divided by mean) of the data prior to regridding, although less restrictive due to the smaller number of profiles and the higher





spatial resolution of the model grid. More specifically, only grid cells with coefficient of variation less than unity were used in the assimilation, while in Cheng et al. (2019) the corresponding threshold was set equal to 0.5.

In addition, in order to avoid spurious values in the assimilation process (e.g. unrealistic high values of extinction coefficient at 532 nm arising from possible misclassification of clouds as aerosols), we discarded LIVAS dust extinction coefficients larger than $10^{-3} \text{m}^{-1}$. Errors in POLIPHON pure-dust extinction coefficient profiles are of the order of 15-25% (Ansmann et al., 2019). In consequence and similarly to Cheng et al. (2019), input error statistics for the data assimilation routine were prescribed as the 20% of the value of the dust extinction coefficient. The corresponding uncertainties in the CALIPSO-based pure-dust product are extensively and in-depth analyzed in Marinou et al. (2017). The number of ingested profiles in the assimilation is shown in the middle panel of Fig. 1. An additional filter was applied to ensure that the 60-meter vertical resolution observations cover at least half of each model layer vertical thickness. Model layers and the corresponding LIVAS observations with less than 50% of vertical coverage were omitted in the observation operator. The remaining observations were averaged and the associated uncertainty was computed assuming a Gaussian correlation length of 1 km in the vertical coordinate for each model layer independently.

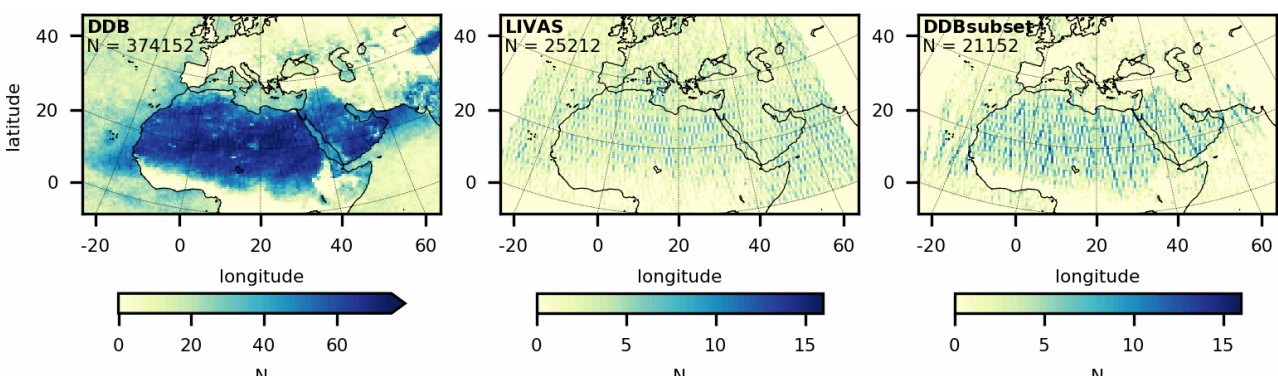

**Figure 1.** Projection on the latitude-longitude map of the number of observations ingested by the assimilation routine. The left panel shows the number of observations ingested by the assimilation routine in the model grid for the dust Deep Blue (DDB) dataset. The middle panel shows the model horizontal distribution of number of observations, counted as the number of LIVAS profiles ingested by the assimilation routine in the model grid. The right panel shows the number of observations ingested by the assimilation routine in the model grid for the DDBsubset dataset. N stands for the number of observations. Please note that the center and the right panels have the same colorbar range.

### 2.1.2 VIIRS Deep Blue dataset

The DOD at 550 nm was extracted from the Deep Blue (DB) Level 2 product of the VIIRS instrument onboard the SUOMI-NPP satellite (Sayer et al., 2018; Hsu et al., 2019). The DB product provides total AOD at 550 nm with a global coverage daily. Along with AOD, the DB product includes a flag with the aerosol type classification of the retrieval (namely dust, smoke, high-altitude smoke, non-smoke fine-dominated, mixed, background and fine-mode dominated), and quality-assurance flags





over ocean and land from one (worst quality) to three (best quality). Hsu et al. (2019) highlight the improvements done in the DB retrieval for dust aerosols, as the optical model was updated with non-spherical dust optical properties.

The standard DB product is AOD. We used only pixels classified as "dust" aerosol type and with a quality assurance flag equal to 3 over ocean and greater than or equal to 2 over land. The resulting DOD dataset was then interpolated to the model grid and assigned an uncertainty of $0.2 \times \text{DOD} + 0.05$ following Sayer et al. (2019). Hereafter we use *DDB* to refer to this filtered dust DB retrieval. The number of DDB retrievals assimilated in this study is shown in the left panel of Fig. 1. We note that DDB is not necessarily a pure-dust AOD and may include contributions of other aerosols types, although dust should be predominant particularly in Northern Africa and the Middle East.

The large swath (30–40 km) of the VIIRS instrument can be a big plus for data assimilation. In contrast, CALIOP has a horizontal footprint of 100 m and a horizontal resolution of 333 m. When comparing the assimilation from both instruments, it is key to understand the role of these differences in spatial coverage. To respond to this fundamental question, we prepared a subset of DDB data, consisting of pixels with DDB retrievals but only where the LIVAS dataset has a valid retrieval in at least one vertical level for every UTC day. This dataset of DOD is called hereafter *DDBsubset*, and the number of DDBsubset retrievals used in the assimilation is shown in the right panel of Fig. 1.

## 2.2 Ground-based observations for evaluation

### 2.2.1 AERONET

We used ground-based measurements for the evaluation. For DOD, we selected the group of AERONET stations (Holben et al., 1998) used in the operational SDS-WAS verification. The list of stations is presented in Section A. We used the AERONET Direct Sun product, version 3, level 2. The AOD was interpolated to 550 nm and we assumed dust to be predominant when Ångström exponents at 440–870 nm were smaller than 0.3. In a similar fashion to the AOD filter used in DDB, this filtered AERONET dataset is not a pure-dust AOD. Nevertheless, it is expected that the main aerosol type in the vertical column of this AERONET filtered dataset is dust, but it can be mixed with other types of aerosols (as for example coarse sea spray).

### 2.2.2 Ground-based lidar CyCARE and Pre-TECT campaigns

The modeled vertical profiles of the dust extinction coefficient at 532 nm were evaluated against measurements from three ground-based lidars of the lidar network PollyNET (Baars et al., 2016; Engelmann et al., 2016) operated in the eastern Mediterranean during the Cyprus Clouds Aerosol and Rain Experiment (CyCARE, Radenz et al., 2017) and the Pre-TECT experiment (http://pre-tect.space.noa.gr/). These lidars were located at Finokalia, Crete, Greece (operated by NOA), Limassol, Cyprus (operated by TROPOS in the frame of CyCARE) and Haifa, Israel (Althausen et al., 2019, operated by TROPOS). With these continuously operating lidars, the vertical profiles of the particle backscatter coefficient at 355 nm, and 532 nm, and 1064 nm, the extinction coefficient at 355 and 532 nm, and the particle depolarization ratio at 355 and 532 nm can be retrieved. Using the obtained particle backscatter coefficient, extinction coefficient and depolarization ratio at 532 nm, the dust-only extinction coefficient can be obtained as described in Mamouri and Ansmann (2014, 2017). With this method, the backscatter-related dust





fraction is calculated based on the known depolarization ratio of pure dust (31%) and the non-dust component (5%). Having the dust-only backscatter coefficient, the dust-only extinction coefficient is determined by the use of the pure dust lidar ratio at 532 nm of 45 sr Mamouri and Ansmann (2016). The non-dust extinction coefficient is calculated similarly depending on

the type of non-dust aerosol. Finally, a consistency check is performed by summing up the dust-only and non-dust extinction profiles and comparing them to the total measured extinction coefficient. More details can be found in Urbanneck (2018).

In contrast to the estimated DOD from AERONET and DB AOD retrievals, which may be affected by other aerosols, the PollyNET measurements can provide pure-dust retrievals (Tesche et al., 2009; Mamouri and Ansmann, 2017). For this reason, we use here the lidar observations from CyCARE and Pre-TECT campaigns in the evaluation of our results.

## 2.3    MONARCH Model

To simulate the dust cycle, we used the Multiscale Online Nonhydrostatic AtmospheRe CHemistry (MONARCH) model (Pérez et al., 2011; Haustein et al., 2012; Jorba et al., 2012; Badia et al., 2017; Di Tomaso et al., 2017; Klose et al., 2021). MONARCH is a fully online integrated system for meso- to global-scale applications developed at the Barcelona Supercomputing Center (BSC). It uses the Nonhydrostatic Multiscale Model on the B-grid (NMMB, Janjic and Gall, 2012) as the meteorological

driver and couples gas-phase and mass-based aerosol modules to describe the life cycle of atmospheric components. It uses the Autosubmit workflow manager (Manubens-Gil et al., 2016), which is particularly useful for efficiently executing assimilation runs. The model provides operational regional mineral dust forecasts at WMO SDS-WAS regional centers (http://sds-was.aemet.es/ ; http://dust.aemet.es/), since 2012 contributes global aerosol forecasts within the multi model ensemble of ICAP initiative (Xian et al., 2019), and will soon integrate the CAMS – Air Quality Regional Production (https://www.regional.

atmosphere.copernicus.eu).

MONARCH contains comprehensive aerosol and chemistry packages, but in this work we only focus and compute mineral dust aerosol. Dust is described using eight particle-size bins within 0.2–20 μm in diameter. The MONARCH dust module is described in detail in Pérez et al. (2011) and Klose et al. (2021). MONARCH offers a diversity of mineral dust emission schemes along with multiple configurations. For this study we computed dust emissions by averaging the emission produced

by the four configurations listed in Table 1. All configurations used a modified version of the dust emission scheme of Ginoux et al. (2001) with modifications described in Klose et al. (2021) that include the use of friction velocity instead of 10 m wind speed, a dust-particle size independent threshold friction velocity for particle entrainment taken as the minimum value of the threshold-function from Shao and Lu (2000), and an emitted size distribution following Kok (2011). The entrainment threshold accounts for soil moisture using the correction from Belly (1964) as described in Ginoux et al. (2001). Areas where dust

emission is allowed are constrained by satellite observations, specifically by the frequency of occurrence (FoO) of the Moderate Resolution Imaging Spectroradiometer (MODIS) DB DOD exceeding 0.2 (Hsu et al., 2004; Ginoux et al., 2012). Dust can be emitted for areas in which FoO > 0.025 (Klose et al., 2021). The four configurations differ with regard to the description of (a) the preferential dust sources used for scaling of the dust emission flux and (b) vegetation and surface roughness effects. Configurations I and II used the MODIS FoO of DOD > 0.2 to scale the dust emission flux obtained with the modified

Ginoux et al. (2001) parameterization. Configurations III and IV used the original topographic source mask from Ginoux et al.





**Table 1.** Summary of the four model configurations used to create the multi-scheme dust emissions.

| Config. | Dust source mask/Scaling | Drag partition | Vegetation/Roughness |
|---------|--------------------------|----------------|----------------------|
| I | MODIS FoO/MODIS FoO | Marticorena and Bergametti (1995) | MODIS LAI/Prigent et al. (2012) |
| II | MODIS FoO/MODIS FoO | Raupach et al. (1993) | Guerschman et al. (2015) |
| III | MODIS FoO/Topo. Sources | Marticorena and Bergametti (1995) | MODIS LAI/Prigent et al. (2012) |
| IV | MODIS FoO/Topo. Sources | Raupach et al. (1993) | Guerschman et al. (2015) |

(2001) as the scaling function. To account for roughness elements on the land-surface, such as vegetation, rocks, or pebbles, configurations I and III used the drag partition parameterization from Marticorena and Bergametti (1995) with corrections from King et al. (2005) in combination with a monthly climatology of MODIS-derived leaf-area index (Myeni et al., 2015) and aerodynamic roughness length data for arid regions from Prigent et al. (2012). In contrast, configurations II and IV utilized

the drag partition parameterization from Raupach et al. (1993) together with a monthly climatology of photosynthetic and non-photosynthetic vegetation cover data from Guerschman et al. (2015) (Klose et al., 2021). The drag partition corrections were applied to the threshold friction velocity for particle entrainment.

## 2.4 Data assimilation

MONARCH is coupled to a Local Ensemble Transform Kalman Filter (LETKF, Hunt et al., 2007). The LETKF implementation

used in this study was built upon the implementation from Miyoshi and Yamane (2007), Schutgens et al. (2010), and Di Tomaso et al. (2017). We used the 4D-LETKF configuration of this code with an assimilation window and forecast length of 24 hours (starting at 0 UTC) and with hourly outputs. In this LETKF implementation, the observations had been compared to the model simulated equivalent observations with collocation in time and space, and then concatenated to construct the observation and the simulated observation (prior) vectors. An ensemble of MONARCH runs is used to estimate the error covariance matrix

of the prior at the observations' times and locations. We used a Gaussian localisation with horizontal scale of 6 grid cells (around 435 km in our model configuration), vertical scale of 1 model level and temporal scale of 12 hours. Unlike Di Tomaso et al. (2017) or Cheng et al. (2019), we computed the analysis every hour instead of only at 0 UTC. With this configuration, the 4D-LETKF acts as a Kalman smoother that effectively localises the influence of the observations in time, and therefore produces better quality analyses throughout the 24-hour assimilation window. This choice is advantageous when the assimilated

observations are temporally distributed along the assimilation window like in the case of LIVAS with daytime and nighttime profiles, or when the observations are more representative of local conditions as is the case of extinction coefficients compared to column-integrated AOD values. In comparison with Di Tomaso et al. (2017) and Cheng et al. (2019), the Kalman Smoother choice described above should provide the same analyses at 0 UTC as the filtering option. Therefore, the analyses-initialised forecasts are identical with both approaches.

A key ingredient in the data assimilation algorithm is the representation of model uncertainty, which in an ensemble-based scheme, like ours, had been derived from the model ensemble. We have generated the MONARCH ensemble by perturbing





dust emissions and by using an ensemble of meteorological initial and boundary conditions analyses (GEFS; Zhou et al., 2017). The model ensemble was constructed with 20 members, concordant with the 20 GEFS ensemble members. The dust emissions in the model (Section 2.3) were perturbed by multiplicative factors that were extracted from a random Gaussian
distribution with a spatial correlation of $250 \, \mathrm{km}$, a mean of unity and a standard deviation of 0.4. In all cases we assumed that the observational errors were uncorrelated, i.e., the observational error covariance matrix was a diagonal matrix.

## 2.5 Experiment description and evaluation

We describe in Section 2.1 the three datasets used in the assimilation: DDB, DDBsubset and LIVAS. Using a fixed configuration of the model and the data assimilation scheme parameters (Sections 2.3 and 2.4), we designed and ran five experiments
assimilating combinations of the three datasets.

The first experiment, named eLIVAS, assimilated the pure-dust extinction coefficient from the regridded LIVAS dataset (Section 2.1.1). A second experiment, named eDDB, assimilated the DOD from the DDB dataset (Section 2.1.2). The third experiment, named eDDBsubset, assimilated the DDBsubset dataset (Section 2.1.2). The fourth, named eLIVAS+DDBsubset, assimilated the LIVAS and DDBsubset datasets, and the fifth, named eLIVAS+DDB, assimilated the LIVAS and DDB datasets.
We ran our data assimilation experiments over a regional domain centred at $20°$ E in longitude and $30°$ N in latitude, which covers North Africa, the Middle East and Europe (e.g. Fig. 2). The model was set up with a rotated latitude-longitude grid with $0.66°$ resolution at the center of the grid, 40 vertical sigma layers, and hourly output of dust concentrations for the 8 size bins. The dust extinction coefficient and DOD were computed with software provided by Gasteiger and Wiegner (2018). We have assumed spheroidal dust particles with the axis ratio distribution shown in Table 2 of Koepke et al. (2015) and the OPAC
refractive index for dust (e.g. $1.53 + 0.0055i$ for $550 \, \mathrm{nm}$) as in Koepke et al. (2015).

We performed the five data assimilation experiments between March and April 2017. A 14-month spinup was run without assimilation to properly initialise the soil moisture content. We also ran a control experiment over the period of study, consisting of an ensemble forecast without data assimilation. For each of the five data assimilation experiments (eLIVAS, eDDB, eDDBsubset, eLIVAS+DDBsubset and eLIVAS+DDB), we obtained two types of simulation outputs: analyses and forecasts.
We produced ensemble forecasts with a forecast length of 24 hours, initialised with the last timestep of the analyses of the day before (at 0 UTC in our 24-hour assimilation window). Forecasts and observations along with their prescribed error were the input for the data assimilation scheme, which computed the 4D mass concentration dust field analyses within the assimilation window. Therefore, for a given day, forecasts can carry the observational information assimilated from the days before, but analyses can, in addition, carry the observational information of that given day. In contrast, the control experiment
omits the assimilation of dust information.

When comparing the model against observations, the model was always collocated in space and time with the valid observations. In the case of ground-based lidar observations, the model is integrated in time over the measurement window. To summarise the comparison between model and observations, we have computed six scores. Five out of the six scores use the model ensemble mean, and one of the scores uses the full ensemble. Given a set of $N$ pairs of model ensemble mean values





$\{m_i\}_{i=1...N}$ and matched observation $\{r_i\}_{i=1...N}$, we use :

$$\text{Mean Bias [MB]} = \frac{1}{N}\sum_{i=1}^{N} m_i - r_i \,,$$

$$\text{Mean Fractional Bias [MFB]} = \frac{2}{N}\sum_{i=1}^{N} \frac{m_i - r_i}{m_i + r_i} \,,$$

$$\text{Pearson correlation coefficient } [\rho] = \frac{\sum_{i=1}^{N}(m_i - \overline{m})(r_i - \overline{r})}{\sqrt{\sum_{i=1}^{N}(m_i - \overline{m})^2}\sqrt{\sum_{i=1}^{N}(r_i - \overline{r})^2}} \,,$$

$$\text{Mean Fractional Error [MFE]} = \frac{2}{N}\sum_{i=1}^{N} \left| \frac{m_i - r_i}{m_i + r_i} \right| \,,$$

$$\text{Root Mean Square Error [RMSE]} = \sqrt{\frac{1}{N}\sum_{i=1}^{N}(m_i - r_i)^2} \,,$$

where $\overline{m}$ and $\overline{r}$ are the average of the model and observations, respectively. We also included the mean over the number of observations of the Continuous Ranked Probability Score (CRPS, Hersbach, 2000, and references therein), which is computed for each observation $r_i$ and model ensemble $m_i^j$, $j = 1\ldots M$ as:

$$\text{CRPS}_i = \int_{-\infty}^{\infty} \left[ P_i(x) - P_{r_i}(x) \right]^2 dx \,,$$

where $P_i$ is the cumulative distribution function of the ensemble, which is approximated empirically by the $M$ ensemble
members, and $P_{r_i}$ the cumulative distribution function of the observation $r_i$, computed as $P_{r_i}(x) = H(x - r_i)$, where $H$ is the Heaviside step function.

## 3 Results and discussion

We first discuss the internal consistency of the data assimilation system in Section 3.1 by comparing analyses and forecasts with the assimilated data. We then present the evaluation against ground-based measurements from Sun photometers and lidars
in Section 3.2.

### 3.1 Consistency and cross-comparison checks with satellite products

We cross-compared the model simulations (control, forecasts and analyses) with the two main assimilated observational datasets. Consistency can be checked when analyses are compared with an observational dataset used for the assimilation (e.g., when DDB DOD are compared to analyses from the eDDB or eLIVAS+DDB experiments). This verification step provides a
sanity check for the data assimilation process. When the datasets are not assimilated (e.g., when DDB DOD are compared to





analyses from the eLIVAS experiment), the comparison is then performed with independent satellite observations. Forecasts are initialised from analyses, thus forecast scores (i.e. error metrics calculated for the forecast fields) can also be considered, up to a certain degree, as an evaluation of the forecast quality, even though the reference observations and the forecast cannot be assumed to be completely independent in this case.

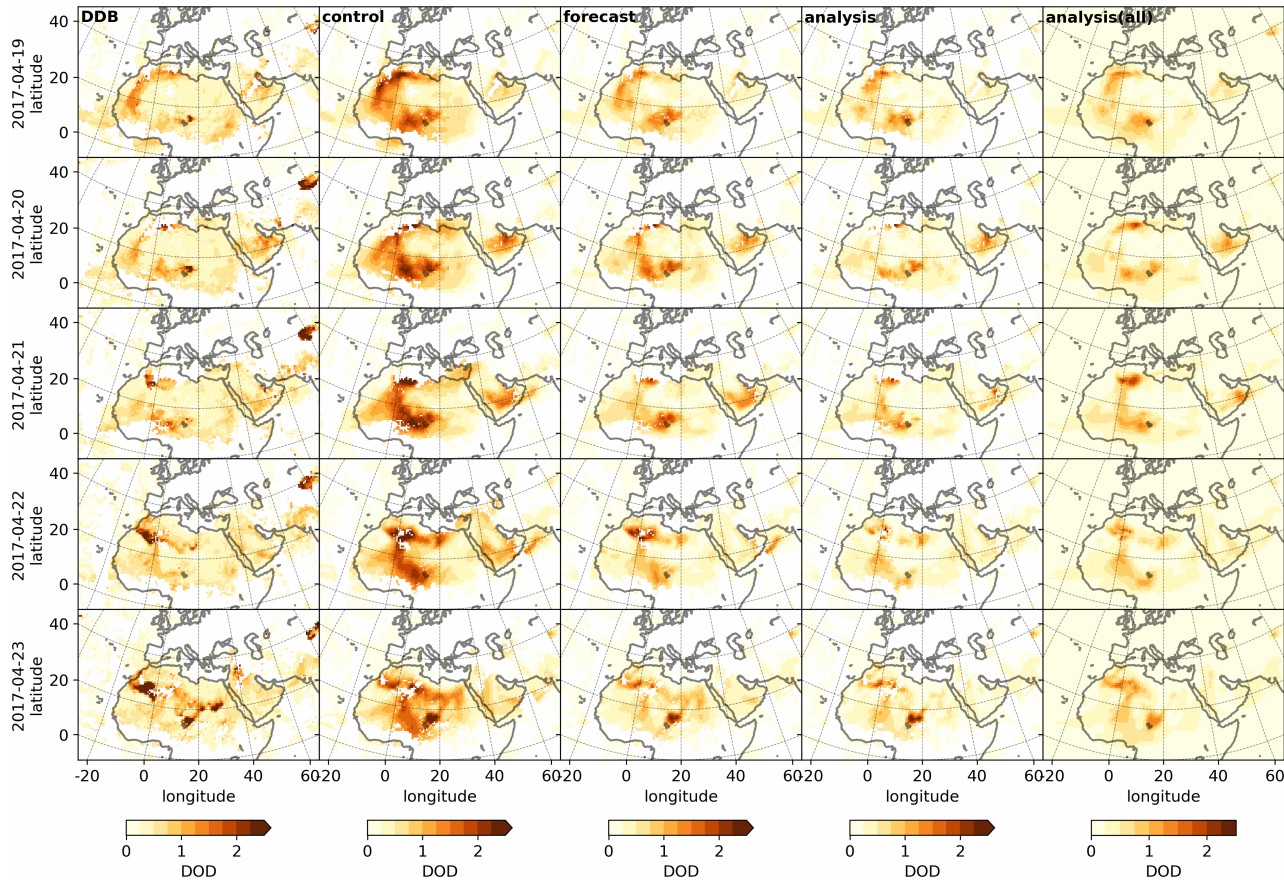

**Figure 2.** DOD from DDB and model simulations between 19 and 23 of April 2017 for the control and eLIVAS experiment. First four columns show the DOD from DDB (left) and three model simulations of DOD collocated with DDB: control experiment, forecast and analysis from the eLIVAS experiment. The last column shows the analyses daily average without collocation with DDB. Each row represents a different day.

A showcase of the eLIVAS experiment is presented in Fig. 2. Here we show the DDB DOD and the control, forecast and analysis DOD for selected days in April 2017, where it is possible to identify a dust plume over the Eastern Mediterranean that was captured by the CyCARE and Pre-TECT campaigns (Section 3.2).

      In contrast to the control run, which overestimates DOD, the analyses and forecasts are in better agreement with DDB both in terms of overall DOD values and spatial distribution. In this experiment, DDB DOD were not assimilated, thus the
qualitative improvement of the analysis compared to the control run indicates that, despite the relatively low spatial coverage



of LIVAS, its assimilation can positively impact the spatial representation in the analysis. Improvements where observations are not available, mainly due to the narrow satellite footprint of CALIOP, are explained by the spatial spread of the observational information through the background error covariance matrix.

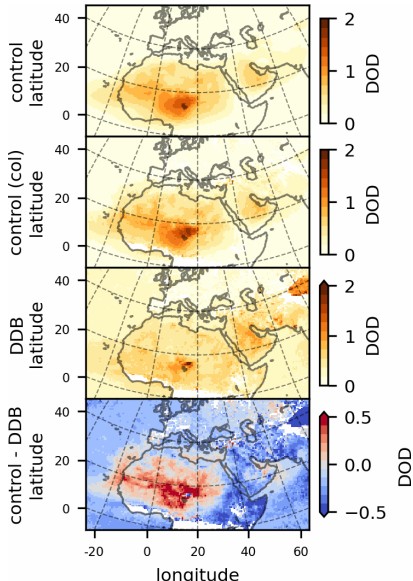

**Figure 3.** Averaged DOD from the control run and DDB during the whole study period. The first panel from the top shows the average of the control run DOD. The second panel shows the average of the control run DOD in the pixels collocated with DDB. The third panel shows the average of the DDB DOD. The fourth panel shows the average difference between the control run and the DDB DOD.

Average maps of DOD are shown in Figures 3 and 4. Compared to the DDB DOD, the control run shows a large over-
estimation of DOD over the Sahara and an underestimation elsewhere (Fig. 3). Figure 3 also shows relatively large values of DOD in the DDB panel over North Atlantic that are not simulated in the control run. The first two rows of Fig. 4 show that the analyses have, in general, lower DOD values than the forecasts, which also have lower DOD values than the control experiment. Third row of Fig. 4 shows the average difference between forecasts and analyses. In this row, there is a common decrease of DOD values close to the Bodélé depression after assimilation and only eDDB and eLIVAS+DDB show increasing
DOD in the eastern part of the domain in the analyses. The impact of the assimilation of observations is larger in the eLIVAS than the eDDBsubset experiments. Averaged differences of the simulations with respect to DDB are shown in the last two rows of Fig. 4. As expected, these differences are smaller in the eDDB and eLIVAS+DDB experiments because the DDB dataset is assimilated in these two experiments.

Figure 5 shows the average values of the assimilated LIVAS pure-dust extinction coefficient profiles compared with collo-
cated model-derived dust extinction profiles for the full domain and the four regions presented in Fig. B1. The model system-atically overestimates the dust extinction coefficient below 7 km in the analyses of experiments excluding LIVAS assimilation and in the forecasts, including experiments with LIVAS assimilation. The altitude of the maximum values and shape of the dust



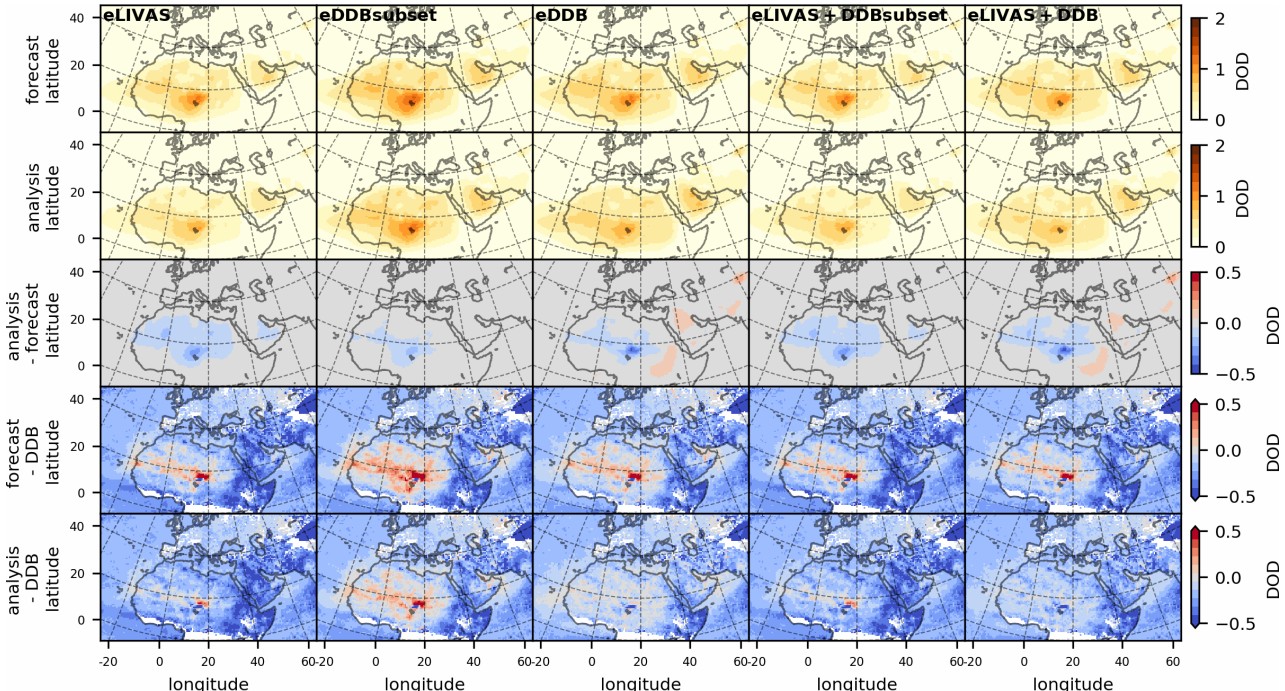

**Figure 4.** Averaged DOD from the experiments and differences with the DDB DOD during the whole study period. Experiments listed in Section 2.5 are represented in columns. The first and second rows show the average DOD for the forecast and analysis experiments respectively. The third row shows the difference between average analyses and forecasts. Collocated differences between the forecasts and the DDB DOD averages are shown in the fourth row. The last row shows the difference between the analyses and the DDB DOD averages.

extinction coefficient is well captured by the model in all regions except the Mediterranean (middle column of Fig. 5), where the mean values are relatively small. Relative changes in the shape of the dust extinction coefficient compared to the control

experiment are shown in the right column of Fig. 5. On average, experiments that assimilate LIVAS and DDB (i.e., eLIVAS, eLIVAS+DDB and eLIVAS+DDBsubset) show stronger decrease in the extinction coefficient than the rest of experiments. Analyses from eLIVAS and eLIVAS+DDB also show large decreases below 3 km of altitude over the Sahara and the Arabian Peninsula. The relative changes in the dust extinction coefficient of analyses and forecasts from eLIVAS and eLIVAS+DDB are larger above 3 km, compared to experiments that exclude LIVAS assimilation. The different shapes of the forecasts and

analyses from eLIVAS and eLIVAS+DDB (right column of Fig. 5) indicate that close to the surface, the dust extinction profile is largely influenced by the model forward simulations (and the associated dust emissions), rather than by the assimilated information. In contrast, in the upper part of the atmosphere the assimilation of LIVAS data adds information to the analyses that is propagated in time by the subsequent forecast cycles. This effect is not as noticeable in experiments excluding LIVAS assimilation. The relatively flat curves associated to eDDB and eDDBsubset show that the shape of simulated dust vertical

profile is mainly propagated from the forward model.

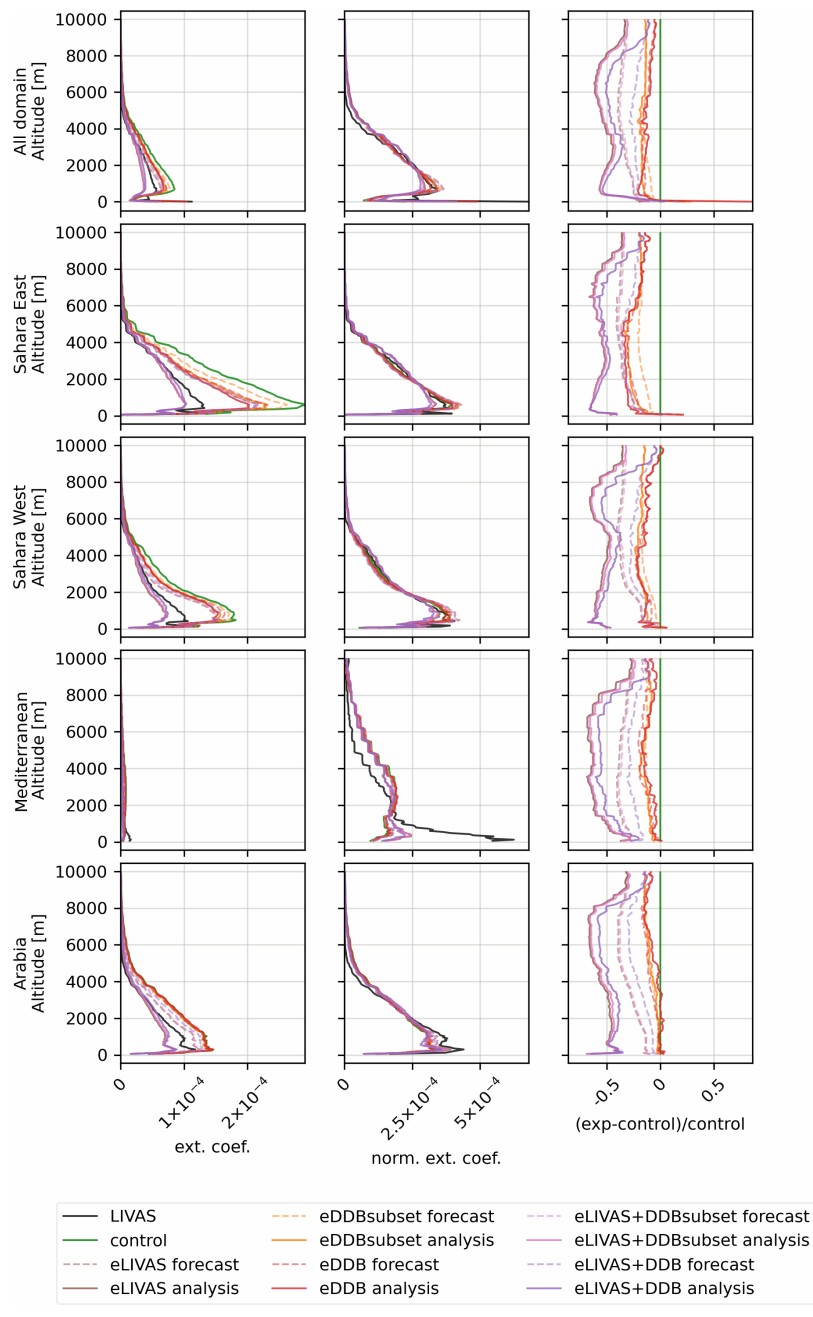

**Figure 5.** Collocated average dust extinction coefficients from model experiments and LIVAS assimilated observations in the 4D domain. The left column shows the mean extinction coefficient profiles in $\mathrm{m}^{-1}$. The middle column shows the mean extinction profiles but normalised such that the vertical integration of each profile in the panel equals one. The right column shows the relative difference between the mean value of each experiment and the control run. Each row represent a different geographical domain defined in Fig. B1, from top to bottom: full domain, East Sahara, West Sahara, Mediterranean and Arabian Peninsula.

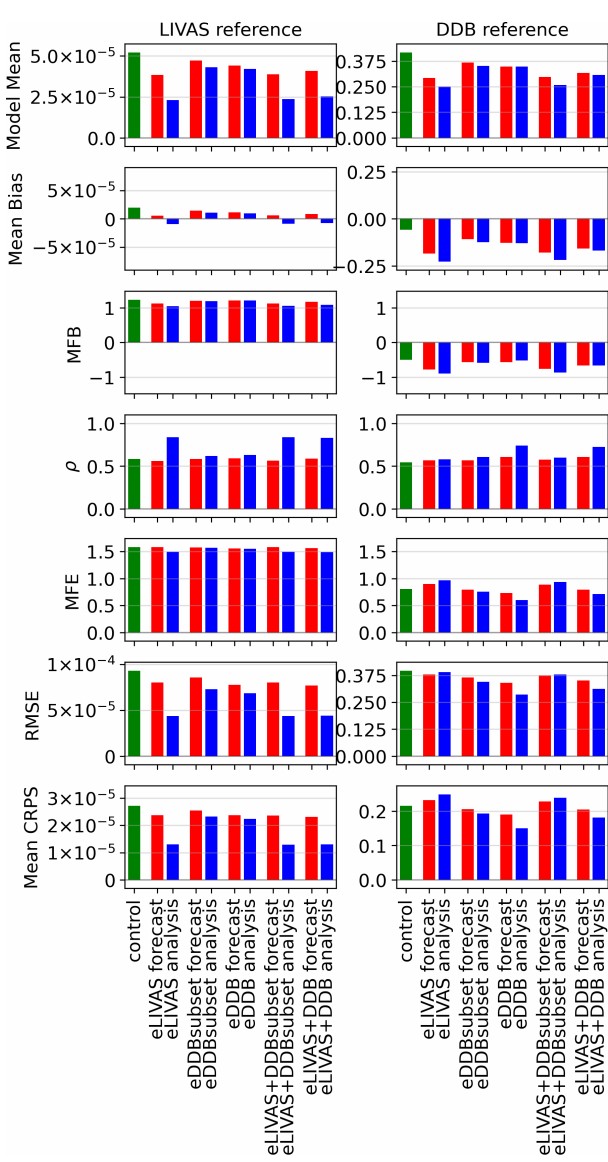

**Figure 6.** Verification scores against LIVAS dust extinction coefficient in the left column, scores against DDB DOD dataset in the right column. Scores of the control run are shown in green, forecasts in red, and analyses in blue. On the left column, panels of Model Mean, Mean Bias, RMSE, and Mean CRPS have units of $\mathrm{m}^{-1}$.





The control run and the five assimilation experiments were also quantitatively evaluated against the LIVAS dust extinction coefficients and the DDB DOD using the scores described in Section 2. When evaluated against LIVAS (left column of Fig. 6), the control run shows a positive mean bias that decreases in the forecasts from all the assimilation experiments. When LIVAS is assimilated either with or without DDB, the analysis is negatively biased. The MFB is positive for all experiments, and its absolute value slightly decreases with the assimilation. The negative mean bias of eLIVAS, eLIVAS+DDB and eLIVAS+DDBsubset analyses suggests that the assimilation tends to decrease rather than increase mass. Small mixing ratios have a smaller spread in the ensemble than the larger ones because the mass mixing ratio (the control vector in the DA) is bounded by zero. This may favour the decrease of mass for large DOD or extinction coefficient values, but not for small values. Amend this behaviour could require, for example, applying non-linear transformations to the control vector and/or the observation operator, which is beyond the scope of this work. As expected, the correlation coefficients obtained for the analyses from experiments assimilating LIVAS are significantly higher than those from the control run. However, the impact of the LIVAS assimilation is limited in the forecasts. Experiments eDDB and eDDBsubset slightly improve the correlation in the analyses but not in the forecasts. The MFE of analyses and forecasts is similar or smaller for the five assimilation experiments than for the control experiment. The RMSE and CRPS decrease similarly for the all the forecasts. As expected, the RMSE and CRPS of eLIVAS, eLIVAS+DDB and eLIVAS+DDBsubset analyses show smaller values. It is also worth noting that the RMSE and CRPS of eDDB and eDDBsubset analyses decrease despite not assimilating LIVAS.

When evaluated against DDB DOD (right column of Fig. 6) the bias is negative for all experiments and products, and even becoming larger with assimilation. The opposite sign in the mean bias of the control run depending on the reference dataset suggests a potential inconsistency between DDB DOD and LIVAS. We note that none of the datasets have been bias-corrected. Inconsistencies may be related to the different quantities retrieved (DOD versus extinction coefficient), each one with its own uncertainties. By construction the DDB DOD (Section 2.1.2) may tend to be positively biased because although the selected scenes are mostly affected by dust, there will be always some contamination by other aerosols in the atmospheric column (see for example the relatively large DDB DOD values over North Atlantic in Fig. 3). On the other side, the low sensitivity of CALIOP to thin layers (Kar et al., 2018) or the signal attenuation in CALIOP measurements when the dust load is large could also play a role in the obtained biases. When comparing with DDB DOD, the correlation coefficient is larger for both forecasts and analyses when the full set of DDB is assimilated (eDDB and eLIVAS+DDB), while the improvement of this score is smaller for eLIVAS, eDDBsubset, eLIVAS+DDBsubset. The analyses show a better correlation coefficient with respect to DDB than the forecasts, and both show a better behaviour than the control run for all the experiments. The three error scores (MFE, RMSE and CRPS) differ across two distinct groups of experiments. These errors increase in eLIVAS and eLIVAS+DDBsubset both for the forecast and analyses with respect to the control run. In contrast, the three other experiments (eDDB, eLIVAS+DDB and eDDBsubset) show a decrease in these error scores both for the forecasts and analyses. The latter is expected as the scores are computed taking the DDB as the reference.

In summary, the results of the cross-comparison checks are consistent with the assimilated observations used in each experiment. Also, when LIVAS is used as the reference dataset, all experiments improve their scores after assimilation both for forecasts and analyses. However, the comparison shows mixed results when the reference dataset is DDB.





## 3.2 Evaluation against ground-based measurements

**Figure 7.** Verification scores against DOD filtered from AERONET AOD observations.

Figure 7 presents the scores for each experiment when evaluated against dust-filtered AOD at 550 nm from AERONET stations (Section A). We acknowledge that the filter used for creating this AERONET DOD dataset (Ångström exponent less than 0.3) can bias our analysis towards large values of DOD. The left column of Fig. 7 shows the scores when all the filtered



observations are taken into account (2681 observations), while the other three columns show the scores in North Africa (1394 observations), the Mediterranean and southern Europe (1029 observations) and the Middle East (258 observations). The list of stations used for each set of scores is listed in Section A and it is based on the list of stations used for operational verification of the SDS-WAS forecasts (http://sds-was.aemet.es/).

     The bias of the control run is positive, which contrasts with the negative bias resulting from the comparison with DDB

(Fig. 6). While the different spatial and temporal localisations used in the comparison may play an important role in this difference, an additional explanation is that the dust filter in the DDB dataset is less conservative and provides on average larger DOD values due at least partly to the presence of other aerosols. The high positive bias of the control run for all AERONET stations decreases in absolute terms in all the experiments after assimilation, consistent with the systematic decrease in the simulated DOD shown in the first row of Fig. 6. Forecast and analyses from experiments where LIVAS was assimilated (eLI-

VAS, eLIVAS+DDBsubset, eLIVAS+DDB) show a stronger negative bias and MFB than experiments where only DDB was assimilated (eDDB, eDDBsubset), notably in the Mediterranean and southern Europe (a subset of 39% of the all AERONET observations used here) and Middle East panels (9% of the all AERONET observations used here).

     The correlation coefficient increases in all assimilation experiements compared with the control for North Africa and the Mediterranean, particularly in the analyses but also in the forecasts. Experiments where the full DDB DOD is assimilated

(eDDB, eLIVAS+DDB) show higher correlations. Over Middle East, with only 258 observations, the correlation coefficient is very low but still positive. Over North Africa, all the experiments show smaller errors (MFE, RMSE, CRPS) in comparison with the control run. With the exception of eLIVAS, all the analyses have smaller errors than the forecasts and –similarly to the correlation coefficient– the experiments where the full DDB DOD was assimilated show better scores in their analyses. Despite the similar horizontal coverage of the assimilated observations, experiment eDDBsubset shows better scores than eLIVAS.

As introduced in Section 2, we used lidar retrievals of pure-dust profiles for the evaluation of the experiments. The evaluation was conducted for the dust event above the eastern Mediterranean between 19 to 24 April 2017, whose extent and dynamics can be observed in the right column of Fig. 2.

     We compared our five experiments with the dust extinction coefficient provided by these lidars. Figure 8 shows the comparison between the lidar measurement in the three sites, the control run, the forecasts and analyses from three experiments

(eLIVAS, eDDB, eLIVAS+DDB), and the AOD from AERONET sites close to the lidar instruments, without filtering by Ångström exponent. Rows 1, 5 and 9 of Fig. 8 show the integrated dust extinction coefficient for lidar measurements and model runs, and the AOD from AERONET close to these stations. The control run is overestimated in the three sites and both analyses and forecast show values closer to the AERONET AOD and the lidar integrated DOD. The three experiments capture the timing and the magnitude of the dust event. Qualitatively, eDDB overestimates the AOD and lidar-integrated measurements,

eLIVAS+DDB is closer to AERONET AOD measurements and eLIVAS is closer to the lidar-integrated DOD. The control run not only overestimates the dust profile but also underestimates the height of the maximum values in the plume (e.g. on Limassol panel, the 21 and 22 of April). For forecasts and analyses, the experiments where LIVAS were assimilated (eLIVAS and eLIVAS+DDB) are able to decrease the dust concentration in the lower layers (below 2.5 km), making the shape of the profiles similar to the observed ones. The eDDB profiles do not show this feature.

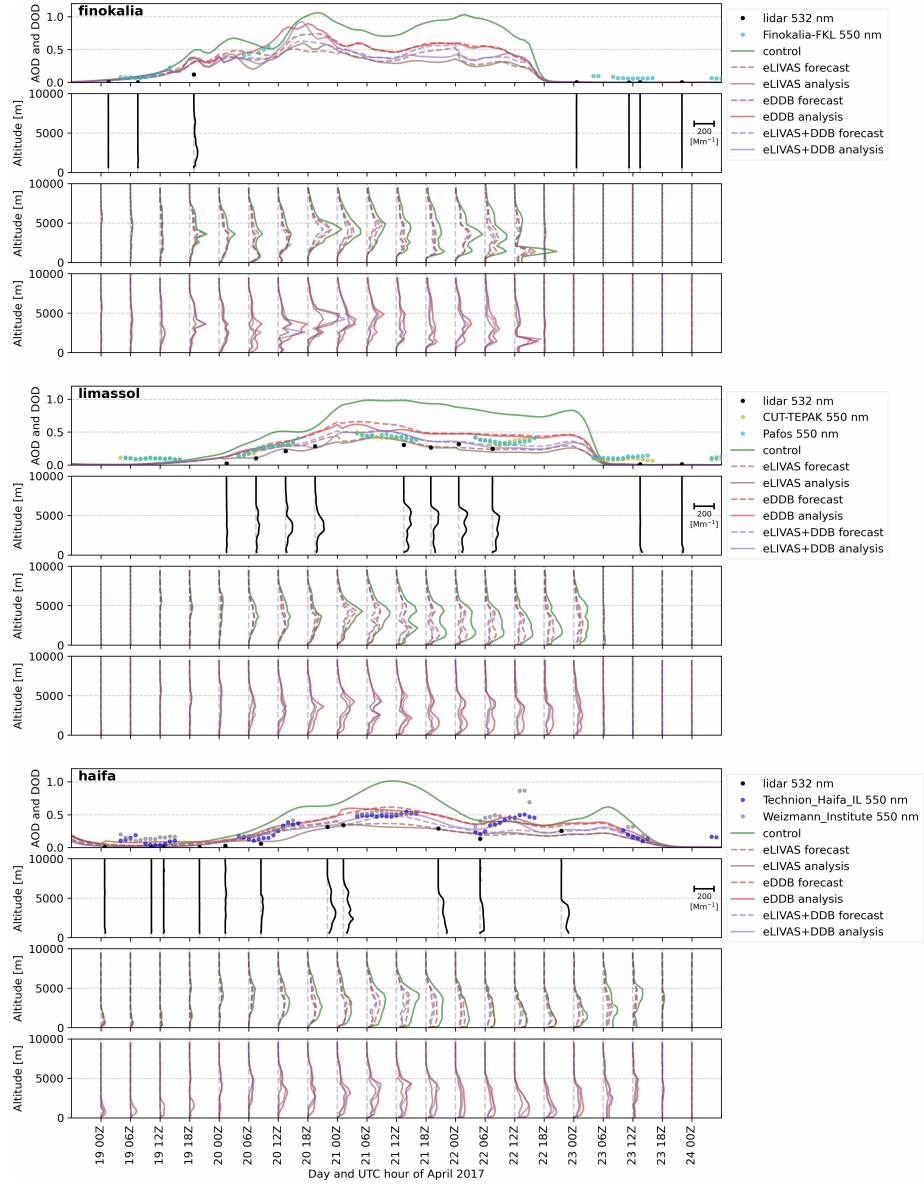

**Figure 8.** Simulated and measured dust extinction coefficient during the CyCARE and Pre-TECT campaigns by PollyNET lidars. The figure contains three groups (for the three measurement sites) with four panels each. The first panel of each group shows the DOD evolution estimated from the lidar measurements (black dots, equal to the vertical integration of the dust extinction coefficient profiles), the AOD without dust filtering for the AERONET station (colored dots), the DOD of the control experiment (green line), the DOD from the forecasts (dashed lines) and analyses (continuous lines) for three selected assimilation experiments. The second row of each group shows the vertical profiles of the measured dust extinction coefficient. The third row of each group shows dust extinction coefficient from the control run (in green) and the forecasts (dashed lines). The fourth row of each group shows the dust extinction coefficient from the analyses. The scale for all the dust extinction coefficient profiles is shown on the right side of the second panel in each group.



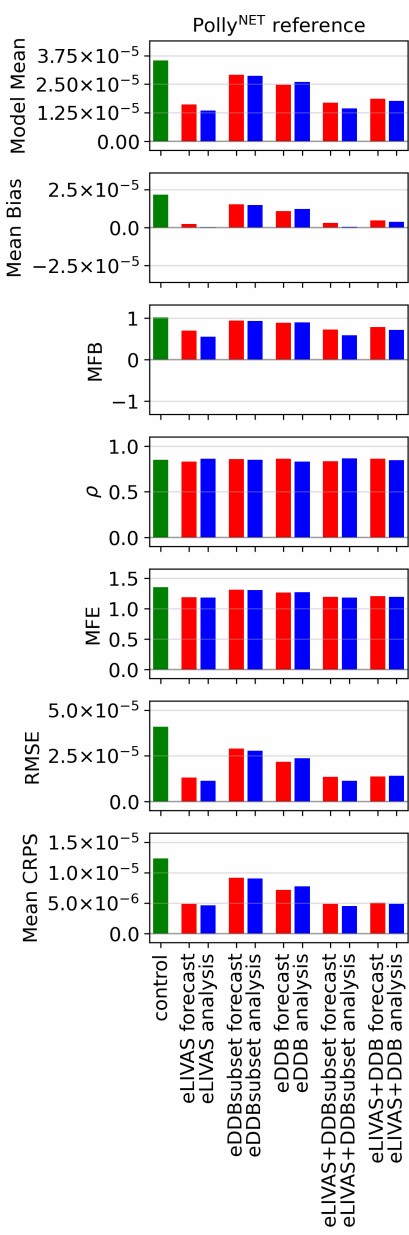

**Figure 9.** Verification scores against ground-based PollyNET lidar dust extinction coefficients from the CyCARE and Pre-TECT campaign. Panels of Model Mean, Mean Bias, RMSE, and Mean CRPS have units of $\mathrm{m}^{-1}$.





The overall quantitative evaluation is shown in Fig. 9. In general terms, both bias scores are smaller for eLIVAS, eLI-
VAS+DBBsubset and eLIVAS+DDB than for eDDB and eDDBsubset. The correlation coefficient is weakly affected by the
assimilation in all experiments and the MFE is slightly smaller for experiments where LIVAS were assimilated. RMSE and
CRPS behave similarly, with improvement for all the experiments compared to the control run, particularly for those where
LIVAS was were assimilated. We also note that in contrast to the evaluation against AERONET DOD, eLIVAS shows better

MFE, RMSE and CRPS scores than eDDBsubset, indicating that the assimilation of pure-dust vertically-resolved observations
can provide a better vertical representation of the dust concentrations than those with only DOD.

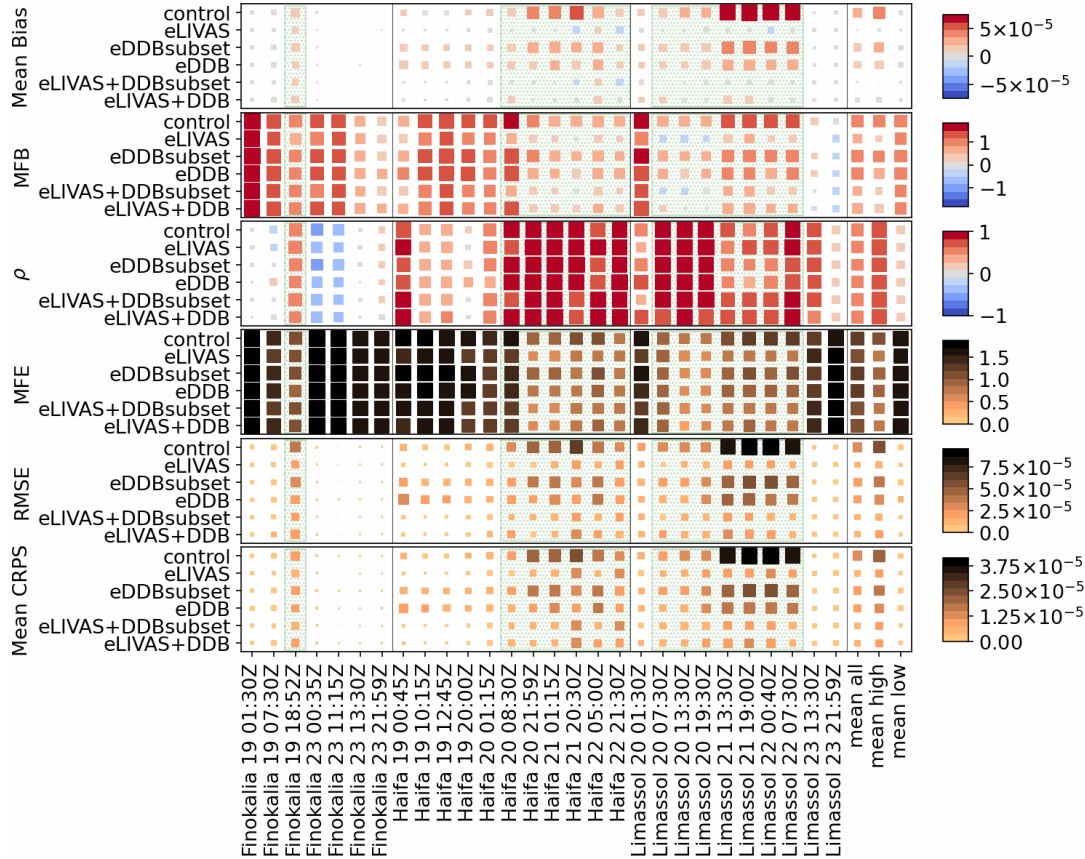

**Figure 10.** Verification scores of the model analyses for the dust extinction coefficient profiles against measurements of PollyNET lidars of
Fig. 8. Mean Bias, RMSE and CRPS have units of $\mathrm{m}^{-1}$. Profiles with high values of extinction are shown with a green shade. The last three
columns show averages of the scores for the non-shaded profiles (mean low), for the shaded profiles (mean high) and for all profiles (mean
all). The areas of the squares are proportional to their values (in colors).

We have computed the evaluation scores also for each of the available profiles (Fig. 8), which are summarised in Fig. 10. The
assimilation performance was split in two groups. The first group is characterised by low values of measured dust extinction
coefficient (non-shaded columns in Fig. 10) where the dust plume cannot be easily identified in Fig. 8. A second group of




profiles corresponds to high dust extinction coefficients (green-shaded columns in Fig. 10). For these profiles, it is possible to visually identify in Fig. 8 the altitude and shape of the dust plume.

In this Section we show two different approaches for computing the scores of a group of vertical profiles. The first approach is to compute the scores by concatenating all the pairs of observed and simulated extinction coefficients, without distinguishing among profiles on the computation. This has been done in Fig. 9. A second approach is to compute the scores for each observed

profile, and then to compute estimators of the statistics of these scores. The last three columns of Fig. 10 follow this approach, showing the average of the scores computed for each profile. In Fig. 10, the *mean all* column shows the average of the scores of all the profiles; the *mean high* column shows the average of the scores of the profiles with strong dust extinction coefficients (i.e., green-shaded columns of this figure); and the *mean low* column shows the average of the scores of the profiles with small values of measured dust extinction coefficients (non-shaded columns in Fig. 10). These last three columns in Fig. 10 are

also shown in Table 2. We note that this methodological difference between both approaches to compute the scores can have noticeable impacts. For example, the *mean all* values of the correlation coefficient and RMSE of Table 2 differ from those in Fig. 9.

For the group of profiles with small values of dust extinction coefficient (*mean low* in Fig. 10) the absolute scores (Mean Bias, RMSE and CRPS) are small because simulated and observed values are also small, but do not improve with assimilation.

Similarly, the normalized scores (MFE and MFB) and the correlation coefficient do not improve. The group of profiles with high dust extinction coefficients (green-shaded columns in Fig. 10) generally show better normalised scores than the group with low dust values. With the exception of the correlation coefficient, all the scores in this group improved after assimilation. In this *mean high* group the Mean Bias drastically decreased when LIVAS were assimilated (eLIVAS, eLIVAS+DDBsubset and eLIVAS+DDB). Similarly, MFB, MFE, RMSE and CRPS improved with LIVAS assimilation. The correlation coefficient

does not improve with assimilation, but the degradation of this score in all experiments with respect to the control run remains below 5% of the control run value. Overall, when dust extinction is large, all analyses improved in all scores with the exception of the correlation coefficient. While improvements are enhanced when LIVAS is assimilated, they are still non-negligible for eDDB and eDDBsubset.

All in all, despite the sparse spatial coverage of LIVAS compared to DDB, this evaluation shows that dust extinction profiles

are best constrained in experiments where LIVAS is assimilated.

### 3.3   Consistency between column and profile assimilation and the role of a narrow satellite footprint

In previous sections we show that experiments assimilating the full DDB dataset (eDDB) yield better scores than those assimilating the DDB subset (eDDBsubset), underlining the importance of the horizontal coverage of the observations in our assimilation. We also show that eLIVAS+DDB produces better scores than eLIVAS when compared to AERONET DOD, and

also that eLIVAS+DDB produces better scores than eDDB when compared to PollyNET. However, we obtained mixed results when comparing eLIVAS and eDDBsubset, which have a similar horizontal coverage: eLIVAS performed better than eDDBsubset when evaluating against PollyNET, but this was not necessarily true in the comparison against DOD. Therefore, we argue





**Figure 11.** Bi-dimensional histograms of the difference between analyses and the control run DOD. Transposed plots in the figure are symmetrical with respect to the 1:1 line. Color scale shows the counts of analysis minus control in a box of $\Delta$DOD $= 0.37$, that is, 151 bins between -2.8 and 2.8. Please note the logarithmic scale of the counts.





**Table 2.** Values of the last three columns of Fig. 10.

| Score | group | control | eLIVAS | eDDBsubset | eDDB | eLIVAS+DDBubset | eLIVAS+DDB |
|---|---|---|---|---|---|---|---|
| Mean Bias ($\times 10^6$) | mean all | 21.91 | -0.29 | 15.08 | 12.30 | 0.56 | 3.97 |
| | mean high | 40.03 | -1.69 | 25.74 | 19.17 | -0.06 | 6.19 |
| | mean low | 3.79 | 1.11 | 4.42 | 5.42 | 1.18 | 1.74 |
| MFB | mean all | 1.02 | 0.55 | 0.93 | 0.90 | 0.58 | 0.71 |
| | mean high | 0.98 | 0.25 | 0.85 | 0.76 | 0.30 | 0.48 |
| | mean low | 1.05 | 0.85 | 1.02 | 1.03 | 0.87 | 0.94 |
| $\rho$ | mean all | 0.51 | 0.50 | 0.49 | 0.50 | 0.51 | 0.52 |
| | mean high | 0.81 | 0.81 | 0.78 | 0.77 | 0.81 | 0.80 |
| | mean low | 0.20 | 0.19 | 0.20 | 0.23 | 0.20 | 0.24 |
| MFE | mean all | 1.35 | 1.18 | 1.30 | 1.27 | 1.17 | 1.19 |
| | mean high | 1.06 | 0.80 | 0.95 | 0.89 | 0.79 | 0.79 |
| | mean low | 1.63 | 1.55 | 1.64 | 1.64 | 1.56 | 1.58 |
| RMSE ($\times 10^6$) | mean all | 29.59 | 9.03 | 22.11 | 19.46 | 9.08 | 11.21 |
| | mean high | 52.35 | 14.54 | 36.13 | 29.16 | 14.66 | 18.43 |
| | mean low | 6.82 | 3.53 | 8.10 | 9.76 | 3.50 | 4.00 |
| Mean CRPS ($\times 10^6$) | mean all | 12.51 | 4.70 | 9.14 | 7.84 | 4.55 | 4.92 |
| | mean high | 22.42 | 8.05 | 15.22 | 11.98 | 7.73 | 8.22 |
| | mean low | 2.59 | 1.35 | 3.05 | 3.70 | 1.37 | 1.62 |

that a direct comparison among experiment analyses can further help elucidating the impact of assimilating vertically-resolved dust observations, along with the impact of the narrow (or wide) field of view of the measurements in our analyses.

Although DDBsubset was designed to have a similar horizontal and temporal coverage than LIVAS, a direct comparison between the eDDBsubset and eLIVAS experiments should also take into account that (i) LIVAS provides direct observational information in the vertical coordinate, while DDBsubset does not; (ii) the vertical influence of LIVAS information is only partial if the column is not complete, in contrast to the DDB DOD that is propagated to the whole column; (iii) DDB only provides data during the afternoon overpass (about 13:30 LT), while LIVAS provides data during afternoon and night overpasses. Nighttime

profiles have better quality, and given the assimilation cycle design and the temporal localisation applied, they should influence the 0 UTC analyses more than daily afternoon, along with the forecast and subsequent analyses.

    It is possible to compare the experiments by inspecting the histograms of differences between the analyses and the control run. We have computed these differences for DOD in Fig. 11 and for dust extinction coefficient in Fig. 12. Figure 11 shows bi-dimensional histograms of the DOD differences for the five experiments. The 1:1 line indicates that respective analyses pro-

duce the same differences with the control run, i.e. they are equal. Points in quadrants I and III indicate that both experiments increase and decrease, respectively, the DOD values at the same locations and times, which is as a sign of consistency. It can be seen that the (eDDB, eLIVAS+DDB) panel shows less deviation with respect to the 1:1 line than the (eLIVAS, eLIVAS+DDB)





case. This indicates that most of the impact of the observations in the eLIVAS+DDB experiments comes from DDB rather than from LIVAS, which is consistent with the scores presented in previous sections. A similar result is found when com-

paring (eLIVAS+DDBsubset, eLIVAS) with (eLIVAS+DDBsubset, eDDBsubset). In this case, eLIVAS+DDBsubset is closer to eLIVAS rather than eDDBsubset. Because the datasets have a similar horizontal coverage, we conclude that either LIVAS add more information to the analyses than the DOD from DDBsubset, or the nighttime overpass of CALIOP has a stronger influence on the 0 UTC analyses, which is also propagated to the forecasts. Similarities between (eLIVAS+DDB, eLIVAS) and (eLIVAS+DDB, eDDBsubset) suggest that the LIVAS assimilation is less important than the DDB assimilation in the

eLIVAS+DDB case, because of the smaller observational coverage. A relatively large spread can be noticed in the (eLIVAS, eDDB) panel and to a lesser extent in the (eLIVAS, eDDB subset) panel.

The spread in the (eDDBsubset, eDDB) panel is associated with the smaller coverage of DDBsubset. In this panel, most values lie around zero in the eDDBsubset axis, which is directly related the reduced amount of assimilated data. A small quantity of values (around the 6% of this panel) are in quadrants II and IV, meaning that the increments with respect to

the control DOD of the eDDBsubset and eDDB analyses are of different sign. A possible explanation is a potentially poor estimation of the terms outside the diagonal of the background error covariance matrix, as they should spread consistently (or at least in the same direction) the DDBsubset observational influence to the remaining pixels covered by the full DDB dataset.

Bi-dimensional histograms of the differences in dust extinction coefficient between analyses and the control run experiment are shown in Fig. 12. In general terms, this figures shows similar, but less clear features than the DOD in Fig. 11. Notable

differences are in the row comparing eDDBsubset with the other experiments, where the values in the panels do not show the clear correlation that appears in the DOD case. This indicates that the shape of the dust profiles in the experiments assimilating LIVAS substantially differ from those assimilating DDB. This is supported also by the eLIVAS+DDB panels, where the larger influence of DDB over LIVAS observations shown for DOD in Fig. 11 is less clear for the extinction coefficient. As we show in Section 2.2.2, the assimilation of LIVAS data (either in eLIVAS or eLIVAS+DDB) can produce more accurate dust

profiles. This demonstrates that the assimilation of vertically-resolved dust extinction coefficients can effectively improve the dust vertical distribution in forecasts and analyses.

## 4    Conclusions

We performed, analysed, and evaluated model experiments assimilating spaceborne dust extinction coefficient profiles and DOD over a two-month period over Northern Africa, the Middle East and Europe. We filtered the AOD observations from

VIIRS DB to obtain a DOD dataset, and we have used for the first time the CALIPSO-based LIVAS pure-dust dataset in a data assimilation framework. In most cases, the assimilation of these products (and their combination) is beneficial for analyses and forecasts.

Experiments that assimilate DDB yield better DOD error scores than those that assimilate only LIVAS when evaluated against AERONET. However, the assimilation of only LIVAS can still achieve significant improvements on these DOD scores.

**Figure 12.** Similar to Fig. 11 but for the dust extinction coefficient. The width of the bins is $4.63 \times 10^{-5}\,\mathrm{m}^{-1}$.





We evaluated the potential improvements in the representation of the dust vertical profile using a handful of high-quality
ground-based lidar pure-dust extinction coefficient measurements performed during the CyCARE and Pre-TECT experimental
campaigns in the Mediterranean. The assimilation of LIVAS leads to a better representation of the dust extinction coefficient
profiles than the assimilation of DDB alone. Jointly assimilating DDB and LIVAS yields the second-best scores for both the
DOD and the dust extinction coefficient profile, which proves their suitability for dust forecast applications.

We have also focused on the limitations of the narrow footprint of LIVAS compared with the large swath of DDB, which
reduces the observational influence on the analyses. Yet, the impact of the vertically-resolved information provided by LIVAS
is significant, and with a similar coverage it produces even a larger impact on the analyses than the assimilation of DOD.

Our findings strongly support the conclusions of Cheng et al. (2019) in that the assimilation of aerosol profiles can improve
their vertical representation in models. We additionally show that the vertical profiles of dust extinction coefficient can be
constrained by assimilating the LIVAS product. We are aware of the limitations of this study due to the limited availability of
ground-based PollyXT lidar measurements. We are looking forward to the publication of ground-based pure-dust lidar datasets
from EARLINET and MPLNET (version 3), that would be very useful for a long term assimilation and evaluation of simulated
dust extinction profiles from model forecasts and analyses. Our work shows the value of space-borne polarization lidars for
improving desert dust forecasts and analyses. As such, future satellite missions with combined extinction and depolarization
capability, such as EarthCARE, are expected not only to further contribute to desert dust research, but also to operational
forecasts if real-time products are made available.

*Code and data availability.* LIVAS pure-dust products are available upon request from Eleni Marinou (elmarinou@noa.gr), Vassilis Amiridis
(vamoir@noa.gr) and Emmanuel Proestakis (proestakis@noa.gr). PollyNET Finokalia data are available upon request from Eleni Marinou
(elmarinou@noa.gr) and Vassilis Amiridis (vamoir@noa.gr). The SUOMI-NPP/VIIRS Deep Blue Aerosol L2 6-Min Swath 6 km was ac-
quired from the Level-1 and Atmosphere Archive & Distribution System (LAADS) Distributed Active Archive Center (DAAC), located
in the Goddard Space Flight Center in Greenbelt, Maryland (https://ladsweb.nascom.nasa.gov/). GEFS data was acquired from the NOAA
National Centers for Environmental Information (https://www.ncdc.noaa.gov/, last access: 25 May 2021). Access to the MONARCH model
code is currently restricted to institutes and collaborators involved in the model development.

**Appendix A: Aeronet sites**

List of AERONET sites used in Section 3.2. The value in parenthesis indicate the number of observations used for each station.
Mediterranean (1029):

AgiaMarina_Xyliatou (2), Aras_de_los_Olmos (7), Badajoz (11), Barcelona (4), Ben_Salem (27), CUT-TEPAK (49), Cabo_da_Roca
(55), Cairo_EMA_2 (70), Carpentras (5), Coruna (15), Eforie (2), Eilat (121), El_Arenosillo (45), Ersa (5), Evora (29),
FORTH_CRETE (12), Finokalia-FKL (19), Galata_Platform (4), Gloria (2), Gozo (19), Granada (34), IMAA_Potenza (1),
IMS-METU-ERDEMLI (29), LAQUILA_Coppito (1), Lamezia_Terme (24), Lampedusa (17), Lecce_University (20), Madrid
(4), Medenine-IRA (84), Messina (4), Modena (1), Montsec (2), Murcia (7), Napoli_CeSMA (4), OHP_OBSERVATOIRE



(5), Palencia (3), Palma_de_Mallorca (11), Rome_Tor_Vergata (9), SEDE_BOKER (99), Tabernas_PSA-DLR (41), Technion_Haifa_IL (49), Tizi_Ouzou (10), Toulon (2), Toulouse_MF (2), Weizmann_Institute (61), Zaragoza (2).

North Africa (1394):

Banizoumbou (123), Capo_Verde (106), Dakar (349), El_Farafra (95), IER_Cinzana (163), Ilorin (47), LAMTO-STATION (50), Saada (80), Santa_Cruz_Tenerife (124), Tamanrasset_INM (257).

Middle East (258):

IASBS (17), KAUST_Campus (97), Masdar_Institute (70), Mezaira (74).

## Appendix B: Regions for LIVAS collocation

We present in Fig. B1 the definition of regions used in Fig. 5.

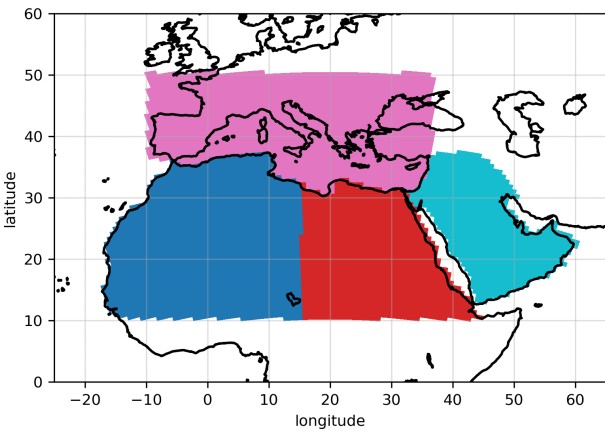

**Figure B1.** Definition of regions of Fig. 5. Mediterranean region is shown in pink, Sahara West in blue, Sahara East in red, and Arabia in cyan.

*Author contributions.* JE designed the assimilation experiments, prepared the DDB dataset, upgraded the MONARCH and data assimilation codes accordingly, performed the simulations, the evaluation, and drafted the manuscript. JE and CPGP designed the study, the manuscript and discussed the main results. EDT, FM and JE developed and maintained the data assimilation code. OJ leads the MONARCH developments with contributions from CPGP, MK, MG, EDT, FM and JE. EM, EP and VA provided the pure-dust LIVAS dataset. CU and HB performed the data analysis of the ground-based lidars. EM, JB, DA and R-EM ensured the ground based lidar instruments performance and data quality during its operation. EM and VA provided the Finokalia dataset. All authors commented on the manuscript.

*Competing interests.* The authors declare that they have no conflict of interest.





*Acknowledgements.* This work has received funding from the European Union's Horizon 2020 research and innovation programme under the Marie Skłodowska-Curie grant agreement No 754433, the European Research Council (grant no. 773051, FRAGMENT) and the
AXA Research Fund. We also acknowledge support from the Ministerio de Ciencia, Innovación y Universidades (MICINN) as part of the BROWNING project RTI2018-099894-B-I00 and NUTRIENT project CGL2017-88911-R, along with PRACE and RES for awarding access to Marenostrum4 based in Spain at the Barcelona Supercomputing Center through the eFRAGMENT2 and AECT-2020-1-0007 projects. MK has received funding from the European Union's Horizon 2020 research and innovation programme under the Marie Skłodowska-Curie grant agreement No. 789630. MK further acknowledges support through the Helmholtz Association's Initiative and Networking Fund (grant
agreement no. VH-NG-1533). VA and EM acknowledge support by the ERC Consolidator Grant 2016 D-TECT: 'Does dust TriboElectrification affect our ClimaTe?' (grant no. 725698). EM acknowledge support from a DLR VO-R young investigator group and the Deutscher Akademischer Austauschdienst (grant no. 57370121). EP has been supported by the project PANhellenic infrastructure for Atmospheric Composition and climatE change (MIS5021516) which is implemented under the Action Reinforcement of the Research and Innovation Infrastructure, funded by the Operational Programme "Competitiveness, Entrepreneurship and Innovation" (NSRF2014–2020) and co-financed
by Greece and the European Union (European Regional Development Fund). This research has been supported by the German-Israeli Foundation for Scientific Research and Development (GIF, grant no.: I-1262-401.10/2014), the European Union's Framework Programme for Research and Innovation, Horizon 2020 (ACTRIS-2, grant no. 654109) and the former European Commission Seventh Framework Programme FP7/2007–2013 (ACTRIS, grant no. 262254) and BACCHUS (grant no. 603445). Lidar observations at Limassol, Cyprus were conducted in collaboration with the Cyprus University of Technology (CUT) and at Haifa, Israel in collaboration with Technion–Israel Institute of Tech-
nology. We therefore grateful acknowledge the support by Yoav Schechner (Technion). We thank EARLINET (https://www.earlinet.org/, last access: 25 May 2021), ACTRIS https://www.actris.eu, last access: 25 May 2021), AERONET (https://aeronet.gsfc.nasa.gov/, last access: 25 May 2021) and AERONET-Europe for the data collection, calibration, processing and dissemination. We thank the PollyNet group for their support during the development and operation of the PollyXT lidars. We thank the NASA/LaRC/ASDC for making available the CALIPSO products which are used to build the LIVAS products.



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
