# Peer review of "Assimilating spaceborne lidar dust extinction improves dust forecasts"

_Atmospheric Chemistry and Physics, 2021_

## Referee Comment (RC2)

In this manuscript, the authors present a study showing the impact of assimilating dust extinction coefficients measured by CALIOP, and Dust Aerosol Optical Depth (DOD), separately and jointly, in a model called MONARCH. The study focuses on two dust source regions during a 2 months period and fields campaigns. The paper is well written and is scientifically interesting. However, there are parts in the article that need clarification. In addition, the paper is a bit long, and the number of case studies makes the paper a little hard to read and follow at times. I am not sure the DODsubset cases are useful in the first part since extinction profiles, and DOD does not contain the same information content; they overcrowd the figures to obtain conclusions that are kind of expected. The paper will gain clarity if you reduce the number of experiments in the main text and move some of them in the appendix if you want to keep them. I would keep the DODsubset experiments for section 3.3.

I recommend accepting it after addressing the following comments:

Title: I would update the title to be more precise. Your study focuses on particular regions of the globe, and your evaluation period is minimal. Therefore, your conclusions are based on comparisons to independent lidar observations measured during fields campaigns that are very limited.

Specific Comments:

Line 71: Please provide references to support both statements: 'Dust is irregularly shaped' and 'most models assume dust...".

Line 72: Since the assumption of optical properties is essential in your assimilation scheme, you need to provide more details on your dust optical properties. How do you derive them? Which spheroidal model are you using? Which refractive indices are you assuming? At least provide references if details come later in the paper.

L85:94: the first paragraph could be largely reduced; you are repeating what has just be said 177:80

L109:115: Please provide more details. On line 100, you specify that LIVAS provides averaged profiles of aerosol properties on a 1x1 degree grid resolution. On Line 109, you specify that you aggregate the profiles of LIVAS to the resolution of the model which is 0.66x0.66 degree grid resolution. It isn't very clear. What about the vertical resolution of retrievals?

L125-127: You mentioned 'cover at least half of each model layer vertical thickness'; what is the vertical resolution of the model? This part is confusing and needs to be more detailed. Is the vertical resolution of CALIOP higher than the model one? How did you adjust the CALIOP profiles to the model resolution? They need to be on a similar vertical resolution.

Figure1: what is the purpose of the figure? The number of retrievals is over which period?

L144: CALIOP has a horizontal footprint of 100m and a horizontal resolution of 333m. From your description of CALIOP LIVAS retrievals, did you use 1x1deg averaged profiles or L2 profiles you averaged at the model horizontal resolution?

L146: Can you provide more details about the collocation between the 2 datasets?

L152: What is SDS-WAS? I think you mean Appendix A? L153-154: Please provide a reference to support your statement?

L189: For the dust emissions, you describe the four configurations listed in Table1, but why averaging 4 configurations? What is the gain of doing that instead of using one particular one? Would you please justify your choice?

Section 2.5: As stated previously, I think five experiments is a lot to follow, considering the gain in conclusion. It would help if you consider moving some experiments to the appendix. The paper is a bit long, and the figures are crowded.

Figure 7: There are too many profiles per plot. The color scale makes it very hard to differentiate the cases. I would also increase the thickness of the lines. I am not sure the experiments with DDBsubset add any value to this plot.

L311: Contrary to the overestimation of the forecast and analyses without eLIVAS assimilation, do the analyses with eLIVAS assimilation seem to underestimate the extinction coefficient?

Figure 8: It is too small, and the color scale again does not help.

L399: "making the shape of the profiles similar to observed ones": the way it is currently in Fig8, it is not easy to tell.

Figure 9: How did you process Figure 9? How many lidar profiles did you use? At Finokalia, it looks like there is no data during the dust event?

L405: "indicating that the assimilation of pure dust vertically resolved observations can provide a better vertical representation than those with only DOD". It was expected since DOD is a column integrated parameter and consequently does not contain any information about the vertical distribution of aerosols. Again, I am not sure of the interest of the eDDBsubset experiment.

L412:414: Why does this line arrive so late?

L479: Section 2.2.2? I think you mean section 3.2.

Conclusion: L485: 'first time': how does CALIPSO-LIVAS extinction data differ from the CALIOP extinction products used in other published papers?

Minor comments:

L33-37: Please update your reference. The study from El Amraoui et al., 2020 https://doi.org/10.5194/amt-13-4645-2020, focuses on assimilating CALIOP extinction coefficients during a dust outbreak.

Please provide full names for each acronym: CALIOP, VIIRS, AERONET, CyCARE, LIVAS, IASI, NOA, TROPOS, ICAP, GEFS.....in the abstract but also in the text. The first time you use the acronym, you should provide the full name.

L182-185: The sentence is too long and needs to be rephrased.

---

## Author Comment (AC1)

**Answer to Reviewer #1**

We appreciate Reviewer #1's positive perspective about our paper. We address below the specific comments. Reviewers' questions and comments are shown in blue fonts, our answers appear in black fonts.

Well done for this really good article either for the quality of the work and the quality of writing. Please find here few comments which I would like to be considerate (it is only typo and colour scales issue).

L72: can you precise here the model used for the spheroidal-shape dust?

We have deleted this sentence in the introduction, but the details and references of the optical model are in L243-245 of the original manuscript.

Figure1: The figure should be placed after the section 2.1.2 as it makes no sense and questioning why the deep blue is here as it is not mentioned in the text yet.

Following suggestions of Reviewer #2, we have removed the eDBBsubset and eLIVAS+DDBsubset experiments from the main text, up to Section 3.3. Therefore, we think that in the revised version of the manuscript this Figure on the number of assimilated observations does not provide interesting information to the reader at his point of the text and it has been deleted.

L152: Section A should be replaced by Appendix A

Done

Figure 5: the colour scale is difficult to read.

Following Reviewer #2 suggestions, we have removed eDBBsubset and eLIVAS+DDBsubset from this Figure, which has improved its clarity. Also, we are now following the colour scale guidelines of colorbrewer.org.

Figure 7: Section A should be replaced by Appendix A

Done.

L 367: Section A should be replaced by Appendix A

Done.

Figure 8: the colour scale is difficult to read.

We have changed the colour scale (as before) and changed the plot appearance.

L404: LIVAS was were assimilated, you should choose between were and was.

Thank you. Done.

**Answer to Reviewer #2**

We thank Reviewer #2 for the insightful review of our paper. We believe the comments have helped improving its clarity. Please find below the item-by-item response to all the comments and suggestions. Reviewers' questions and comments are shown in blue fonts, our answers appear in black fonts.

In this manuscript, the authors present a study showing the impact of assimilating dust extinction coefficients measured by CALIOP, and Dust Aerosol Optical Depth (DOD), separately and jointly, in a model called MONARCH. The study focuses on two dust source regions during a 2 months period and fields campaigns. The paper is well written and is scientifically interesting. However, there are parts in the article that need clarification. In addition, the paper is a bit long, and the number of case studies makes the paper a little hard to read and follow at times. I am not sure the DODsubset cases are useful in the first part since extinction profiles, and DOD does not contain the same information content; they overcrowd the figures to obtain conclusions that are kind of expected. The paper will gain clarity if you reduce the number of experiments in the main text and move some of them in the appendix if you want to keep them. I would keep the DODsubset experiments for section 3.3.

We agree that the DOD and dust extinction coefficient do not contain the same information, and that is precisely the interest of combining and comparing the assimilation of both retrievals. We are also aware that not only the information content of each observation is different, but also the spatial and temporal coverage of the assimilated retrievals is different between the DOD and dust extinction coefficient datasets. The eDDBsubset experiment is indeed designed to help us disentangle these two fundamental differences between the two types of observations.

We agree with Reviewer #2 that the original version of the manuscript can be hard to follow since it contained the description and results of five experiments. Therefore, we have largely followed Reviewer #2's recommendations and moved the description and results relative to the eDDBsubset to Section 3.3 and partly to a new appendix. While up to Section 3.3, the eDDBsubset experiments might be of less interest to the reader, the side-by-side comparison and scores of the eDDBsubset experiments with the full eDDB and eLIVAS experiments provides the rationale and motivation of Section 3.3. In a compromise between the "crowded" figures that the reviewer is well pointing out and the readability of Section 3.3, we have removed the eDDBsubset experiments from the figures and the main text of the paper up to Section 3.3. However, in order to link and provide the motivation and rationale of Section 3.3, we have added a new appendix where we reproduce the skill scores figures with the addition of the eDBBsubset experiments.

I recommend accepting it after addressing the following comments:

Title: I would update the title to be more precise. Your study focuses on particular regions of the globe, and your evaluation period is minimal. Therefore, your conclusions are based on comparisons to independent lidar observations measured during fields campaigns that are very limited.

Our study shows robust improvements in dust forecasts (not only analyses) when comparing with independent ground based Raman lidars, an aspect that has little precedent in the literature. Note that most of the cited previous studies on aerosol lidar DA (L33-34 of the original manuscript) provide verifications of the analyses only, and not forecast-initialised analyses. This is something that we think should be highlighted in the title of our paper. However, we understand the concern of Reviewer #2 and we agree that our study does not cover all the world's regions and periods. All in

all, we have maintained the original title but nuanced in a way that acknowledges to a large extent the reviewer's suggestion.

We have change the title to: "Assimilating spaceborne lidar dust extinction can improve dust forecasts"; replacing the original "Assimilating spaceborne lidar dust extinction improves dust forecasts".

Specific Comments:

Line 71: Please provide references to support both statements: 'Dust is irregularly shaped' and 'most models assume dust…".

We have deleted this sentence in the introduction, because it does not provide interesting information at this stage of the article. However, in situ measurements show that dust is highly aspherical (Kandler et al., 2011, and references therein). Moreover, a recent study (Huang et al., 2020) that compiled measurements of dust shape worldwide concluded that the ratio of dust's longest to shortest dimensions is ∼5 on average.

Line 72: Since the assumption of optical properties is essential in your assimilation scheme, you need to provide more details on your dust optical properties. How do you derive them? Which spheroidal model are you using? Which refractive indices are you assuming? At least provide references if details come later in the paper.

The aerosol optical model is referenced in L243-245 of the submitted manuscript, in the section where the model and observation operator is described :

"The dust extinction coefficient and DOD were computed with software provided by Gasteiger and Wiegner (2018). We have assumed spheroidal dust particles with the axis ratio distribution shown in Table 2 of Koepke et al. (2015) and the OPAC refractive index for dust (e.g. 1.53 + 0.0055i for 550nm) as in Koepke et al. (2015)."

We think that the details of these computations are not appropriate for the article's introduction section. Rather than expanding this sentence in the introduction, we have deleted it in order to shorten this section and kept all the relevant details about the used optical properties Sect 2.5 of the revised manuscript.

L85:94: the first paragraph could be largely reduced; you are repeating what has just be said l77:80

Thanks for the suggestion. Indeed, L85-94 of the submitted manuscript presents with more details what was outlined in L77-80. We have removed the repeated information from L77-80 and kept the first paragraph of Section 2 unchanged.

L109:115: Please provide more details. On line 100, you specify that LIVAS provides averaged profiles of aerosol properties on a 1x1 degree grid resolution. On Line 109, you specify that you aggregate the profiles of LIVAS to the resolution of the model which is 0.66x0.66 degree grid resolution. It isn't very clear. What about the vertical resolution of retrievals?

The reviewer is right about a possible confusion between the 1x1 degree and the aggregation of profiles that we have made in this study.

The standard LIVAS product is produced at a 1x1 degree resolution, but, as we state in L104 of the original manuscript, we use the Level 2 (CALIOP-equivalent) overpass resolution: "… at CALIPSO per-orbit level, used in the present study."

We modified the text to clarify that the standard LIVAS product is produced at 1x1 degree resolution, but we do not use the 1 degree resolution product in this work. Instead, we use the LIVAS data at the CALIOP per-orbit overpass resolution, that is, the set of profiles that are used to create the 1 x 1 degree resolution standard LIVAS product by the LIVAS developers.

Regarding the vertical resolution, it is stated in the original manuscript in L125 (60 m). We have also included it in the revised paragraph of the LIVAS description for clarity.

We think that the level of detail of the LIVAS product description is more than enough for the purpose of this work. Providing more details (that can be found in the cited references) would not add useful information to the reader, as this work is not focused on the development of dust remote sensing products.

L125-127: You mentioned 'cover at least half of each model layer vertical thickness'; what is the vertical resolution of the model? This part is confusing and needs to be more detailed. Is the vertical resolution of CALIOP higher than the model one? How did you adjust the CALIOP profiles to the model resolution? They need to be on a similar vertical resolution.

The model itself does not have fixed number of vertical layers. The user, in the desired configuration, prescribes the appropriate number of layer and hybrid pressure-sigma coefficients for their application. Therefore, the definition of the quantity of model layers is written in Section 2.5 ("Experiment description and evaluation") in L242 of the original manuscript.

In the vertical hybrid pressure-sigma coordinate, the thickness of these layers may vary with the altitude, pressure and topography. However, we have added in the text rough estimates for the thickness of some key layers (surface at sea level, maximum thickness in the troposphere, and 10km of altitude). The thickness of the layers can be easily identified in the "stairs-like" profile shapes of Figure 5 of the original manuscript (Figure 4 in the revised version).

The detailed procedure to "adjust CALIOP profiles to the model resolution" is described in L109-129 of the original manuscript. Filters aside, the last sentence of these paragraphs reads : "… The remaining observations were averaged and the associated uncertainty was computed assuming a Gaussian correlation length of 1 km in the vertical coordinate for each model layer independently." In summary, the assimilated observations at the model vertical resolution are computed by averaging the filtered LIVAS 60m profiles that overlaps the model layers altitude, weighted by their overlapping fraction in the model layer.

Figure1: what is the purpose of the figure? The number of retrievals is over which period?

We have removed this figure from the manuscript. It contains 13 figures, that would increase to 14 with the addition of the figures with eDBBsubset scores in the Appendix. Since we have removed the references to eDDBsubset in most of the main text, this Figure on the number of observations assimilated does not provide anymore interesting information to the reader.

L144: CALIOP has a horizontal footprint of 100m and a horizontal resolution of 333m. From your description of CALIOP LIVAS retrievals, did you use 1x1deg averaged profiles or L2 profiles you averaged at the model horizontal resolution?

This is related to the reviewer's comment above about L 109-115 of the submitted manuscript. We used L2 profiles. We hope that the changes made in the revised version text done in L103-104 clarify this point.

L146: Can you provide more details about the collocation between the 2 datasets?

Temporal collocation is done with a daily resolution and spatial collocation is done by collocating valid retrievals of both DDB and LIVAS profiles. We have modified this paragraph to include more details on the collocation procedure. Now it reads:

"… we prepared a subset of DDB data, called hereafter DDBsubset, that contains the (regridded) DOD from DDB collocated with LIVAS. This collocation is done at a daily resolution and in the horizontal model grid (which is the same horizontal grid of DBB and LIVAS after the processing described in the previous paragraphs). For each UTC day, we create a bi-dimensional binary mask whose values are set to valid only when the LIVAS dataset has a valid retrieval in at least one vertical level, for that UTC day. This daily mask is applied to DDB to create DDBsubset."

L152: What is SDS-WAS? I think you mean Appendix A?

SDS-WAS is the Sand and Dust Storm Warning Advisory and Assessment System of the World Meteorological Organization. It is defined in L54-55 of the original manuscript.

L153-154: Please provide a reference to support your statement?

Following the reviewer's comment, we have added the reference to Basart et al. (2009), where the threshold of 0.3 is explicit in the article. More in general, low values of Angstrom exponent for mineral dust aerosols have been reported for decades (Eck et al. 1999, Schuster et al. 2006).

L189: For the dust emissions, you describe the four configurations listed in Table1, but why averaging 4 configurations? What is the gain of doing that instead of using one particular one? Would you please justify your choice?

The MONARCH model includes a large number of dust emission schemes and configurations (see Klose et al., 2021). In principle, we had no strong reason to prefer one dust emission configuration over the others. We had the option to tune and choose the best performance emission scheme for this model configuration, resolution and period. Our tests indicate that each of these configurations can show very good performance in specific regions, and worse performance in others, depending on the season of the year. Our intention in this work is not to produce the best analyses or forecasts, but rather to show insights on the assimilation of lidar and DOD products. A fine tuning of the dust emission processes for this work would provide a false guidance on the necessary developments for the upgrade of the MONARCH model and the WMO Barcelona Dust Forecast Center; and a false impression in the paper conclusions, as the operational forecasts cannot be tuned a posteriori for specific periods.

With a pragmatic approach, and given the uncertainties in the dust emission modelling processes, we have chosen the equally weighted average of these four configurations. An advantage of choosing four configurations is that the spread on spatial and temporal distribution of emissions could be larger than using only one configuration (by also having more pixels with emissions); and this is helpful for the ensemble data assimilation approach as it offers a more realistic representation of the uncertainty in the emission process. We have, therefore, replaced these lines of text:

"For this study we computed dust emissions by averaging the emission produced by the four configurations listed in Table 1"

by the following (L195-199 of the revised manuscript):

"As shown in Klose et al. (2021), the emission scheme and their specific configuration has a strong impact on the spatial and temporal behaviour of the simulated dust. Because we aim at showing the impact on the forecast of assimilating two different types of observations and not to show the best of the forecasts, we preferred to avoid fine-tuning and cherry-picking the best performant emission scheme and configuration for our study case. Instead, we computed dust emissions by averaging the emissions produced by the four configurations listed in Table 1."

Section 2.5: As stated previously, I think five experiments is a lot to follow, considering the gain in conclusion. It would help if you consider moving some experiments to the appendix. The paper is a bit long, and the figures are crowded.

Thanks for the suggestion. As stated earlier, we have removed eDDBsubset from the main text, until Section 3.3.

Figure 7: There are too many profiles per plot. The color scale makes it very hard to differentiate the cases. I would also increase the thickness of the lines. I am not sure the experiments with DDBsubset add any value to this plot.

We thank the reviewer for this comment. We have changed the colours and removed eDBBsubset from this figure, and we hope that now the figure reads better.

L311: Contrary to the overestimation of the forecast and analyses without eLIVAS assimilation, do the analyses with eLIVAS assimilation seem to underestimate the extinction coefficient?

Yes. It has been added to the text.

Figure 8: It is too small, and the color scale again does not help.

We have changed the colour scale and increased the font size and width of the lines.

L399: "making the shape of the profiles similar to observed ones": the way it is currently in Fig8, it is not easy to tell.

We have updated Figure 8 (Figure 7 in the revised manuscript) to help the reader to identify this aspect more easily.

Figure 9: How did you process Figure 9? How many lidar profiles did you use? At Finokalia, it looks like there is no data during the dust event?

We think that this comment is related with reviewer's comment below about L412:414. At Finokalia there are lidar data just before the dust event. There are no AERONET data during the event neither because it was cloudy.

L405: "indicating that the assimilation of pure dust vertically resolved observations can provide a better vertical representation than those with only DOD". It was expected since DOD is a column integrated parameter and consequently does not contain any information about the vertical distribution of aerosols. Again, I am not sure of the interest of the eDDBsubset experiment.

We partially agree with the reviewer. However, it is important to note that the improvement in the forecast skills by assimilating lidar aerosol observations is not trivial. The improvements in the analyses are more easily achieved. As state before, please note that most of the cited previous studies of aerosol lidar DA (L33-34 of the original manuscript) provide verifications of the analyses only, and not forecast-initialised analysis. We think that evaluating the forecasts initialised by analyses of lidar data is important for the aerosol prediction community (as it was indicated in Section 4.2.3 of Benedetti et al., 2018).

There are several possible reasons on why the assimilation of lidar extinction coefficient can degrade aerosol forecasts. First, the extinction coefficient retrievals have a smaller temporal and spatial scale than AOD. This implies that the impact of each of these observations should be more localised and also, that the balance of errors in the data assimilation system have to be carefully checked (they can have larger representativeness error, for example). If the errors in the data assimilation system are not well balanced, it is very easy to produce an over-fitting of the assimilated data, that would probably degrade the forecasts.

When it is assimilated jointly with AOD, both retrievals can be inconsistent and produce a degradation of the forecasts (with respect a pure-AOD assimilation for example). In addition, we agree that the DOD itself does not contain information in the vertical, but when it is ingested in our data assimilation scheme (with a flow-dependant background error covariance matrix) it is able to change the vertical profile of aerosols, as shown in the red lines of the third column of Figure 5 of the original manuscript (Figure 4 of the revised version). This is indicating that, even though each independent DOD retrieval is profile-agnostic for the assimilation, the set of DOD measurements (with space and time dimensions) can provide information to the vertical representation of the model dust.

Finally, as we have removed eDBBsubset from most of the manuscript text, the sentence in L405 was removed too.

L412:414: Why does this line arrive so late?

We have moved the first part of this paragraph to the text where Figure 9 (now Figure 8) is discussed, while keeping the rest of the text in the discussion of Figure 10 (now Figure 9) .

We expect that this also answer the reviewers' comment on Figure 9 above.

L479: Section 2.2.2? I think you mean section 3.2.

Absolutely. Corrected.

Conclusion: L485: 'first time': how does CALIPSO-LIVAS extinction data differ from the CALIOP extinction products used in other published papers?

The description of LIVAS is in Section 2.1.1. LIVAS uses CALIOP measurement to decouple the dust contribution from the rest of aerosols. CALIOP products provide aerosol typing, but not pure-dust profiles. This is a fundamental difference.

Minor comments:

L33-37: Please update your reference. The study from El Amraoui et al., 2020 https://doi.org/10.5194/amt-13-4645-2020 , focuses on assimilating CALIOP extinction coefficients during a dust outbreak.

Done.

Please provide full names for each acronym: CALIOP, VIIRS, AERONET, CyCARE, LIVAS, IASI, NOA, TROPOS, ICAP, GEFS……in the abstract but also in the text. The first time you use the acronym, you should provide the full name.

Done.

L182-185: The sentence is too long and needs to be rephrased.

We have split the sentence in two.

References:

Basart, S., Pérez, C., Cuevas, E., Baldasano, J. M., and Gobbi, G. P.: Aerosol characterization in Northern Africa, Northeastern Atlantic, Mediterranean Basin and Middle East from direct-sun AERONET observations, Atmospheric Chemistry and Physics, 9, 8265–8282, https://doi.org/10.5194/acp-9-8265-2009, 2009.

Benedetti, A., Reid, J. S., Knippertz, P., Marsham, J. H., Di Giuseppe, F., Rémy, S., Basart, S., Boucher, O., Brooks, I. M., Menut, L., Mona, L., Laj, P., Pappalardo, G., Wiedensohler, A., Baklanov, A., Brooks, M., Colarco, P. R., Cuevas, E., da Silva, A., Escribano, J., Flemming, J., Huneeus, N., Jorba, O., Kazadzis, S., Kinne, S., Popp, T., Quinn, P. K., Sekiyama, T. T., Tanaka, T., and Terradellas, E.: Status and future of numerical atmospheric aerosol prediction with a focus on data requirements, Atmos. Chem. Phys., 18, 10615–10643, https://doi.org/10.5194/acp-18-10615-2018, 2018.

Eck, T. F., Holben, B. N., Reid, J. S., Dubovik, O., Smirnov, A., O'Neill, N. T., Slutsker, I., and Kinne, S. (1999), Wavelength dependence of the optical depth of biomass burning, urban, and desert dust aerosols, *J. Geophys. Res.,* 104( D24), 31333– 31349, doi:10.1029/1999JD900923

Gasteiger, J. and Wiegner, M.: MOPSMAP v1.0: a versatile tool for the modeling of aerosol optical properties, Geoscientific Model Development, 11, 2739–2762, https://doi.org/10.5194/gmd-11-2739-2018, 2018.

Huang, Y., Kok, J. F., Kandler, K., Lindqvist, H., Nousiainen, T., Sakai, T., Adebiyi, A., and Jokinen, O. (2020). Climate models and remote sensing retrievals neglect substantial desert dust asphericity. Geophysical Research Letters, 47(6), 1–11. https://doi.org/10.1029/2019GL086592

Kandler, K., Lieke, K., Benker, N., Emmel, C., Küpper, M., Müller-Ebert, D., Ebert, M., Scheuvens, D., Schladitz, A., Schütz L., and Weinbruch, S. (2011). Electron microscopy of particles collected at Praia, Cape Verde, during the Saharan Mineral Dust Experiment: Particle chemistry, shape, mixing state and complex refractive index. Tellus B: Chemical and Physical Meteorology, 63(4), 475–496. https://doi.org/10.1111/j.1600-0889.2011.00550.x

Klose, M., Jorba, O., Gonçalves Ageitos, M., Escribano, J., Dawson, M. L., Obiso, V., Di Tomaso, E., Basart, S., Montané Pinto, G., Macchia, F., Ginoux, P., Guerschman, J., Prigent, C., Huang, Y., Kok, J. F., Miller, R. L., and Pérez García-Pando, C.: Mineral dust cycle in the Multiscale Online Nonhydrostatic AtmospheRe CHemistry model (MONARCH) Version 2.0, Geosci. Model Dev., 14, 6403–6444, https://doi.org/10.5194/gmd-14-6403-2021, 2021.

Koepke, P., Gasteiger, J., and Hess, M.: Technical Note: Optical properties of desert aerosol with non-spherical mineral particles: data incorporated to OPAC, Atmospheric Chemistry and Physics, 15, 5947–5956, https://doi.org/10.5194/acp-15-5947-2015, 2015.

Schuster, G. L., Dubovik, O., and Holben, B. N. (2006), Angstrom exponent and bimodal aerosol size distributions, *J. Geophys. Res.*, 111, D07207, doi:10.1029/2005JD006328.